

# The implementation of dust mineralogy in COSMO5.05-MUSCAT

Sofía Gómez Maqueo Anaya[1], Dietrich Althausen[1], Matthias Faust[1], Holger Baars[1], Bernd Heinold[1], Julian Hofer[1], Ina Tegen[1], Albert Ansmann[1], Ronny Engelmann[1], Annett Skupin[1], Birgit Heese[1], and Kerstin Schepanski[2]

[1]Leibniz Institute for Tropospheric Research (TROPOS), Leipzig, Germany
[2]Free University of Berlin, Berlin, Germany

**Correspondence:** Sofía Gómez Maqueo Anaya (maqueo@tropos.de)

**Abstract.** Mineral dust aerosols are composed from a complex assemblage of various minerals depending on the region they originated. Giving the different mineral composition of desert dust aerosols, different physico-chemical properties and therefore varying climate effects are expected.

Despite the known regional variations in mineral composition, chemical transport models typically assume that mineral dust
aerosols have uniform composition. This study adds, for the first time, mineralogical information to the mineral dust emission scheme used in the chemical transport model COSMO-MUSCAT. We provide a detailed description of the implementation of the mineralogical database, GMINER (Nickovic et al., 2012), together with a specific set of physical parametrizations in the model's mineral dust emission module. These changes lead to a general improvement of the model performance when comparing the simulated mineral dust aerosols with measurements over the Sahara Desert region for January - February 2022 .
The simulated mineral dust aerosol vertical distribution is tested by a comparison with aerosol lidar measurements from the lidar system Polly[XT], located at Cape Verde. For a lofted mineral dust aerosol layer on the 2 February 5:00 UTC the lidar retrievals yield on a dust mass concentration peak of $156\,\mu g/m^3$ while the model calculates the mineral dust peak at $136\,\mu g/m^3$. The results highlight the possibility of using the model with resolved mineral dust composition for interpretation of the lidar measurements since higher absorption the UV-VIS wavelength is correlated to particles having higher hematite
content. Additionally, the comparison with in-situ mineralogical measurements of dust aerosol particles show how important they are, but also that more of them are needed for model evaluation.

## 1  Introduction

Mineral dust aerosols impact climate in multiple ways. They interact with both shortwave and longwave radiation modifying the atmospheric radiation balance (Kok et al., 2023; Mahowald et al., 2010). Furthermore, they modify cloud properties by
acting as cloud condensation nuclei or as ice nucleating particles (Atkinson et al., 2013; Hoose et al., 2008; Chatziparaschos et al., 2023). Since mineral dust aerosols are a complex assemblage of various minerals with their distinct physicochemical properties (Formenti et al., 2011), differences of their atmospheric radiative impact will arise as a consequence of distinct mineral content. For example, whether the presence of mineral dust aerosols in the atmosphere will result in cooling or warming of the atmospheric column depends on the mineral dust aerosol composition, its size distribution and on surface albedo



(Balkanski et al., 2007). In particular, minerals containing iron oxides are correlated to a distinct interaction with radiation in the UV/VIS wavelength region that carries the possibility of modifying the interaction between the mineral dust aerosols and the atmosphere radiation budget (Kok et al., 2023; Li et al., 2021; Baldo et al., 2020; Balkanski et al., 2007; Jickells et al., 2005). Especially, the feldspar mineral is active as ice-nucleating particles under mixed-phase cloud conditions (Chatziparaschos et al., 2023; Atkinson et al., 2013). Considering that mineral dust aerosols are identified as the atmosphere dominant

natural ice nucleating aerosol (Hoose et al., 2008), recognizing mineral composition of dust aerosol particles is crucial for a more accurate representation of the atmosphere-mineral dust interaction. Moreover, mineralogical composition can be used to identify dust source regions. Soils that commonly produce mineral dust aerosols, so-called mineral dust productive soils, are classified by their distinct combination of mineral fractions content (Formenti et al., 2011, 2014; Scheuvens et al., 2013).

Despite known regional variations in the mineral composition, chemical transport models typically assume that mineral

dust aerosols have uniform composition (Perlwitz et al., 2015b). Therefore, the introduction of mineralogical information in chemical transport models is required for a more accurate simulation of the interaction between mineral dust aerosols and radiation, and ice nucleation, and ultimately weather forecasts and climate simulations (Nickovic et al., 2012; Perlwitz et al., 2015b). The addition of mineral composition to mineral dust simulations becomes even more crucial when considering that dust emissions are likely to change in the future in response to anthropogenic changes in climate and consequent changes in the

natural vegetation pattern and in land use. Dust emissions from new sources due to human activity will have as a consequence a change in the overall composition of airborne dust and therefore modify their atmospheric interactions (Harrison et al., 2001).

Adding mineralogical fractions to mineral dust aerosol simulations in chemical transport models requires specifying the geographic distribution of minerals in mineral dust productive soils at a high spatial resolution. Claquin et al. (1999) proposed that the soil mineral fractions are approximately related to the soil type; taking into account the size distribution, the chemistry

and the color of the soil. They derived an average surface mineralogy that can be inferred for each soil unit of the arid soil. Nickovic et al. (2012) extended this approach by including additional measurements, soil types and minerals. This study presents the inclusion of Nickovic et al. (2012) high-resolution mineralogical database, GMINER (Global Mineral Database), into the mineral dust emission scheme considered in the regional chemical transport model, COSMO-MUSCAT (COSMO: COnsortium for Small-scale Modelling; MUSCAT: MultiScale Chemistry Aerosol Transport Model). Our project focuses on

the region of the Sahara Desert, being of crucial importance for the globe atmospheric mineral dust load (Prospero et al., 2002). In addition, the ability of COSMO-MUSCAT to accurately reflect the mineral dust life cycle (i.e., emission, transport and deposition) has been thoroughly validated for the region, as shown by Heinold et al. (2011); Schepanski et al. (2017, 2009), among others. This study is the first time that soil mineralogical information is included in the mineral dust emission scheme for the regional chemical transport model COSMO-MUSCAT, and it is the first time mineralogical information is included in

the mineral dust life cycle for a regional model.

Studies that have added mineral information to mineral dust aerosol simulations including Scanza et al. (2015) and Li et al. (2021) who focused on the direct radiative forcing of mineral dust aerosols by using the Community Atmosphere Model 4 (CAM4) and the Community Atmosphere Model 5 (CAM5) models. On a similar research line, Journet et al. (2014) use their updated high-resolution mineralogical database to infer mineral dust single scattering albedo via the amount of mineral




contained iron oxide. The model used for that study is the LMDz-INCA (LMDz:Laboratoire de Météorologie Dynamique; INCA: INteraction with Chemistry and Aerosols) which is a coupled General Circulation model (LMDz) with a Chemistry-Aerosol model (INCA). A modelling effort comparing the mineralogical databases of Journet et al. (2014) and Nickovic et al. (2012) with mineral measurements on a global model can be found in Gonçalves Ageitos et al. (2023), who used MONARCH (Multiscale Online Nonhydristatic AtmospheRe CHemistry model) for assessing the sensitivity to the different mineralogy

databases and their effect on the calculation on the direct radiative effect of dust.

    The study performed by Perlwitz et al. (2015b) applies a different approach where by focusing on predicting regional variations of aerosol mineral composition as a function of particle size by considering that the aerosol mineral content may differ from that of the mineral dust productive soil. This has been implemented in the NASA Goddard Institute for Space Studies (GISS) Earth System ModelE2 in which the dust emission scheme follows the work of Kok (2011). Other modelling

studies implementing mineralogy databases with the purpose of researching the link between minerals and their ice nucleating properties involve the works from Hoose et al. (2008), who used the aerosol-climate modelling system ECHAM5-HAM (Global general circulation model ECHAM5 with a coupled aerosol chemistry and microphysics package HAM) model, and the work from Atkinson et al. (2013) who implement mineralogical information in a two-moment microphysical aerosol scheme called GLOMAP (GLObal Model of Aerosol Processes), that runs within the TOMCAT (Toulouse Off-line Model of Chemistry And

Transport) chemical transport model, in order to specifically explore the role of feldspar for ice nucleation in mixed-phase clouds. Furthermore, the study from Chatziparaschos et al. (2023) used the global 3-D chemistry transport model TM4-ECPL in order to investigate the role of feldspar and quartz as ice nucleating particles.

    This paper is structured as follows: the methodology section starts with a general description of the COSMO-MUSCAT model (Section 2.1), followed by the definition of the simulated mineral dust life cycle and a comprehensive explanation of

the modifications done to the dust emission parameterization (Section 2.2), including the implementation of the mineralogical information (Section 2.3). The methodology section closes with describing a particular model setup together with the model validation methods and data acquisition (Section 2.5). On the results section, the aerosol optical thickness (AOT) comparisons between model results and both ground-based and space-borne remote sensing are presented first (Section 3.1). Secondly, mineral mass concentration maps are compared together with mineral mass measurements (Section 3.2). The section ends with

a vertical profile comparison for Mindelo, São Vicente, in the Cape Verde archipelago (Section 3.3). In the conclusions section future research opportunities in regard with minerals optical properties are highlighted.

## 2   Methodology

### 2.1   Model description

The COSMO-MUSCAT model is a mesoscale atmospheric model system that consists of two online-coupled models. COSMO

is a regional forecast model from the German Weather Service (DWD) which solves the governing equations throughout a terrain-following coordinate system (Baldauf et al., 2011). MUSCAT, driven by the meteorological model, is the chemical transport model that calculates the atmospheric advective transport of aerosols taking into account time-dependent mass balance



equations (Heinold et al., 2011; Wolke et al., 2012). Mineral dust aerosols transported in MUSCAT are considered as passive tracers meaning that dust aerosol particles are not chemically aged or involved in any other chemical reactions.

The capability of COSMO-MUSCAT of simulating the Saharan mineral dust aerosol distribution has been validated in the scope of several studies. This includes the study by Heinold et al. (2011) who simulated the transport of mineral dust and biomass-burning smoke from the Sahara Desert towards the Atlantic, whereas the study by Tegen et al. (2013) modelled dust source activation for over two years. Both studies compared and validated their simulations with both ground-based and space-borne remote sensing measurements and found good agreement with the observations. A further validation is shown by

Schepanski et al. (2009) who validated model results for both winter and summer distinctive meteorological conditions. In Schepanski et al. (2016) a more recent version of the model is used for simulating North African dust transport towards the Mediterranean, and it is approved by the comparison of dust loading data from both space-borne and ground-based remote sensing measurements. The conclusion from this study is that the model is trustworthy for simulating the mineral dust life cycle within the atmospheric circulation. A similar conclusion is reached by the comparison done in Schepanski et al. (2017),

where the model was used for simulating the Saharan dust outflow towards the North Atlantic.

The atmospheric mineral dust life cycle is simulated following physical parameterizations defined in MUSCAT. These parameterizations depend on the COSMO meteorological and hydrological fields that get updated for each advection time step. These physical parameterizations include: (1) Dust emission following Tegen et al. (2002) updated by some modifications to consider specific soil mineralogical fractions. The modifications are thoroughly explained in Sections 2.2 and 2.3. (2) Aerosol

transport, which is simulated through solving a third-order upwind scheme via temporal integration (Wolke and Knoth, 2000), and (3) aerosol removal, which includes both dry and wet deposition processes. The computation of dry deposition follows Seinfeld and Pandis (2016) and Zhang et al. (2001), while wet deposition, which comprises both rain-out and wash-out is parameterized following Berge (1997) and Jakobsen et al. (1997). A full description of the deposition scheme can be found in Heinold et al. (2011).

As part of the simulating mineral dust life cycle, COMO-MUSCAT can take into account the aerosol-radiation interactions (Helmert et al., 2007). For the purposes of this study, the aerosol radiative feedback is not considered.

## 2.2 Dust emission

Dust emission is a non-linear process that occurs when the surface wind speed exceeds a threshold. This threshold is based on the wind speed needed to produce a momentum change via vertical shear stress at the soil surface that would be enough to create

particle mobilization. Such particle mobilization is based on the soil friction velocity, which in turn depends on the soil particle size distribution as well as on surface conditions like vegetation cover and soil moisture. The threshold friction velocities for each particle size distribution are calculated following the parametrizations developed by Marticorena and Bergametti (1995).

In order to calculate the soil threshold fiction velocities, first a calculation of the wind produced surface friction velocity is needed. The wind derived surface friction velocity, in turn, depends on the surface wind speed and on the assumption of

a neutral atmospheric stratification and adiabatic conditions. Hence, the logarithmic layer profile theory (Priestley, 1959) is



applied for its calculation, that being

$$u^* = \frac{U(z)\kappa}{\ln\frac{z}{z_0}}. \tag{1}$$

Here a modification of the formula is implemented following Darmenova et al. (2009). They recommend that, when using the emission scheme developed by Marticorena and Bergametti (1995), a fixed value for the local aerodynamic roughness length,

$z_0$, can be used in the logarithmic wind profile implementation (Eq. 1). The fixed value is set to a representative value for bare (desert) surfaces, $z_0$=0.01 m. Specifically, we use the 10 m wind speed as the surface wind velocity ($U(z)$), since this calculation represents the model wind speed which is the closest to the surface, $z$ is the thickness of the surface layer, and $\kappa$ represents the Von Kármán constant.

     Dust emissions can then only take place where the particle size dependent friction velocity is high enough to mobilize dust

particles and where the soil conditions would allow such emissions. When the particle mobilization starts, the first movement is parameterized as a horizontal particle flux,

$$G = \frac{\rho_a}{g} \cdot u_*^3 \cdot \sum_i \left[ \left( 1 + \frac{u_{*tr}(Dp_i)}{u_*} \right) \left( 1 - \frac{u_{*tr}^2(Dp_i)}{u_*^2} \right) \cdot s_i \right] \text{ for } u_* \geq u_{*tr} \tag{2}$$

where $G$ represents the horizontal particle flux, $\rho_a$ is the air density, $g$ denotes the gravitational acceleration, $u_*$ is the surface friction velocity, $u_{*tr}$ is the threshold friction velocity that depends on the soil particle diameter ($D_{pi}$). $i$ represents the different

size fractions of which the diameter depends upon as well as the relative surface area covered by that size fraction, $s_i$. For our setup, we have 196 particle size classes ($i$). The condition for Eq. 2 stresses the fact that mobilization of dust will only happen when the surface friction velocity surpasses the particle size dependent threshold friction velocity. Processes such as saltation and bombardment (Marticorena and Bergametti, 1995) are already considered for the derivation of Eq. 2. After these processes, the flux that is allowed to be transported in the atmosphere is parameterized as a vertical particle flux,

$F = \alpha \cdot A_{eff} \cdot G \cdot (1 - A_{snow}) \cdot I_\theta$                                                 (3)

where $F$ represents the vertical particle flux and $\alpha$ is the sandblasting efficiency which is a ratio between the horizontal fluxes and vertical fluxes and is prescribed for each soil type, considering its percentage of clay, silt, and sand composition of the soil per grid cell as implemented in Tegen et al. (2002). $A_{eff}$ is the erodible area which depends on vegetation cover and roughness length, $A_{snow}$ is the part of $A_{eff}$ covered by snow and $I_\theta$ represents the influence of soil moisture (Fécan et al., 1999). When

a certain amount of vegetation and soil moisture are present, they will have the effect of suppressing mineral dust emission.

     Vertical dust particle fluxes are then transported into the atmosphere depending on the particle size distribution. The particle fluxes are distributed over five independent size classes described in Table 1. As illustrated by Eq. 2 and 3, soil conditions as particle size distribution, soil moisture, vegetation, and roughness length have to be considered as part of the dust emission scheme. Some of the soil conditions inhibit mineral dust emissions, such as soil moisture and vegetation cover. For soil mois-

ture, a threshold is calculated for each grid cell; if the moisture is below this threshold, dust emission is allowed to continue unaltered, but if it is above it, then a linear relationship is followed that suppresses dust emission. The approach follows the parameterization developed by Fécan et al. (1999) and the volumetric soil water content data is obtained for that of the first



| Bin name | Size range | $Q_{ext,550nm}$ |
|----------|------------|-----------------|
| BIN 01 | 0.2 - 1 μm | 1.684 |
| BIN 03 | 1 - 3 μm | 3.165 |
| BIN 09 | 3 - 9 μm | 2.352 |
| BIN 26 | 9 - 26 μm | 2.145 |
| BIN 80 | 26 - 80 μm | 2.071 |

**Table 1.** Description of MUSCAT five independent size classes for mineral dust aerosol transport. Their size limits are indicated together with the dimensionless dust extinction efficiency at 550 nm, $Q_{ext,550nm}$ (Sokolik and Toon, 1996), which is used as a first approximation to calculate atmospheric aerosol optical thickness (AOT) for validation purposes.

soil layer as given by the ERA5 land hourly data (Muñoz Sabater and Service, 2019). Since vegetation cover greatly influences mineral dust emission (Marticorena et al., 1997, Tegen et al., 2002), a limit is set for the fraction of vegetation cover per grid

cell that will suppress dust emission. This fraction is set to 0.5 for deserts and a linear relationship is established between the vegetation cover fraction and the suppression of mineral dust emission following Tegen et al. (2002). Soil vegetation cover information is obtained from the Copernicus Global Land Service, where satellite retrieved FCOVER is the fraction of vegetation cover that corresponds to the fraction of ground covered by green vegetation (Fuster et al., 2020). Soil roughness length is a potential mineral dust emission inhibitor as quantified in the calculation of the particle dependent threshold friction velocity

in Marticorena and Bergametti (1995) mineral dust emission scheme. The aerodynamic roughness length dataset developed by Prigent et al. (2005) is used here as the soil roughness length considered for the mineral dust emission calculations. The particle size distribution for soils, described by the most commonly used size populations, i.e., clay, silt and sand, is obtained from the SoilGrids database (Poggio et al., 2021). In addition, a dust source activation frequency map derived from MSG-SEVIRI IR-channels as described in Schepanski et al. (2007) is used to verify dust emitting areas.

**2.3 Mineralogy implementation**

The relation between soil particle size distribution and type with its fractional mineral abundance is described by Nickovic et al. (2012), where the high-resolution mineralogical database, GMINER, is introduced. GMINER exclusively considers mineral fractions for the mineral dust productive soils and it is specifically aimed for atmospheric dust modelling. The mineralogical database follows the Claquin et al. (1999) procedure which establishes the identification of mineral dust productive soils

following the FAO74 classification (FAO-UNESCO, 1974) and provides information on soil populations of clay and silt sized particles. Effective fractions of minerals in soils are determined by combining soil texture classes and applying modifications derived from modelling approaches. GMINER is consequently a database that establishes the relationship between different mineral dust-productive soil types and the following minerals: quartz, feldspar, calcite, gypsum, illite, kaolinite, smectite, hematite and phosphorus. Mineral fractions are distributed over clay and silt particle size population, where clay is defined as

particles with sizes less than 2 μm and silt is classified as particles sizes between 2 μm and 50 μm. In such a framework, certain minerals are considered as having contributions to both clay and silt size populations as is the case with quartz, while other





minerals just have contributing fractions to either clay or silt sizes. For example, illite contributes only to the soil clay particle size population and feldspar is only considered as part of the silt particle size population. Noteworthy is that the particle size value of 2 μm, which acts as the division between mineral contents in clay and silt sized particles, is thought as arbitrary and was used for the creation of the database as a first approximation (Nickovic et al., 2012). For the present study that value remains unchanged.

The mineralogical fractions of dust aerosol particles are assumed to be the same as these mineral fractions of the soil source. This implies that the changes on particle size distribution due to the saltation and sandblasting processes does not affect the mineral distribution, even though the general mineral dust particle size distribution is modified during emission (Journet et al., 2014; Marticorena and Bergametti, 1995; Perlwitz et al., 2015a). This approach is chosen because the GMINER data set already partially considers the modification of the mineralogical composition of mineral dust aerosols due to the size distribution modification during emission. By only taking into the account the soil mineralogical composition of the particle classes that would be emitted, that being, silt and clay sizes. The two smallest size classes are considered light enough to remain suspended in the atmosphere for several days.

The approach by Nickovic et al. (2012) introduces various sources of uncertainty. For instance, the relation between mineral composition and soil type is derived from a sparse amount of measurements, and they are generalized for soil types. That means that regional variations in mineral content for a particular soil type are not considered. Furthermore, some measurements are based on wet sedimentation techniques that disturb the original soil composition, breaking aggregates and therefore replacing them with smaller particles. That causes a larger allocation of mineral fractions to clay sized populations than could exist in undisturbed soil. This could cause large differences in soil size distributions during emission (Journet et al., 2014).

Measurements of the different particle size distributions between source soil composition and suspended mineral dust are sparse. Such measurements are not straightforward and the amounts of mineral dust aerosol collected downwind of the source are small (Journet et al., 2014). Nevertheless, Caquineau et al. (1998) and Lafon et al. (2004) have shown that the relative proportions of clay sized minerals in the mineral dust aerosol are close to those of the mineral dust productive soils and are conserved during transport. Results were later supported by Perlwitz et al. (2015b) who found that the particle composition of the clay sized emitted minerals is identical to that of the fully dispersed soil as given by Claquin et al. (1999). On the other hand, mineral distribution in silt sizes appears to not be the same as the distribution found on the soil. Perlwitz et al. (2015b) suggest that this is because of the wet sieving technique used to measure the mineral content on the soil. They take into account a size modification of the emitted dust via Kok (2011)'s brittle fragmentation theory that is independent of the size distribution of the soil. This suggests that indeed the silt sized mineral particles have a different size distribution than they would have when setting the aerosol fraction to be equal to the fraction found in the parent soil. However, the exact physical parameters that rule the size distribution changes depend on the mineral dust emission scheme used, which can vary substantially. Consequently, several studies (Atkinson et al., 2013; Hoose et al., 2008; Journet et al., 2014) have assumed that the particle size distribution of the emitted minerals resembles that of the mineral dust productive soil.

Figure 1 shows a representation of the partitioning of the dust particles regarding their mineral content both for the source soil size distribution and the aerosols size distribution for a particular grid cell located in mid-western Mali. Mineralogy for



dust particles are derived from a simulated dust emission flux that took place on 27 January 2022 21:00 UTC. The emitted mass fractions are normalized so that within each size bin, the sum over all minerals is unity. The normalization is chosen to show the minerals proportions in each size class to both the soil composition and on the mineral dust aerosol size classes. The

vertical emission size classes are described in Section 2.2 and Table1.

For each size bin, emission fluxes in kg/m$^2$s are calculated taking into account the meteorological conditions of that location and time. Mineral fractions are appointed following the mineral dust productive soil mineral fraction. For the mineral calculation, clay and silt sized minerals divide the total mineral dust emission for each size class. As such, for BIN03 (1-3 µm), a mixture of clay and silt size minerals divides the size class total emission, and for BIN80 (26-80 µm) minerals fractions repre-

sent only a small portion of the overall mineral dust emission content. This is due to the cut on the mineralogical dataset where the classification ends with particles of diameters of 50 µm, therefore, all the particles that are bigger than that in the model are considered as not having any kind of mineral information.

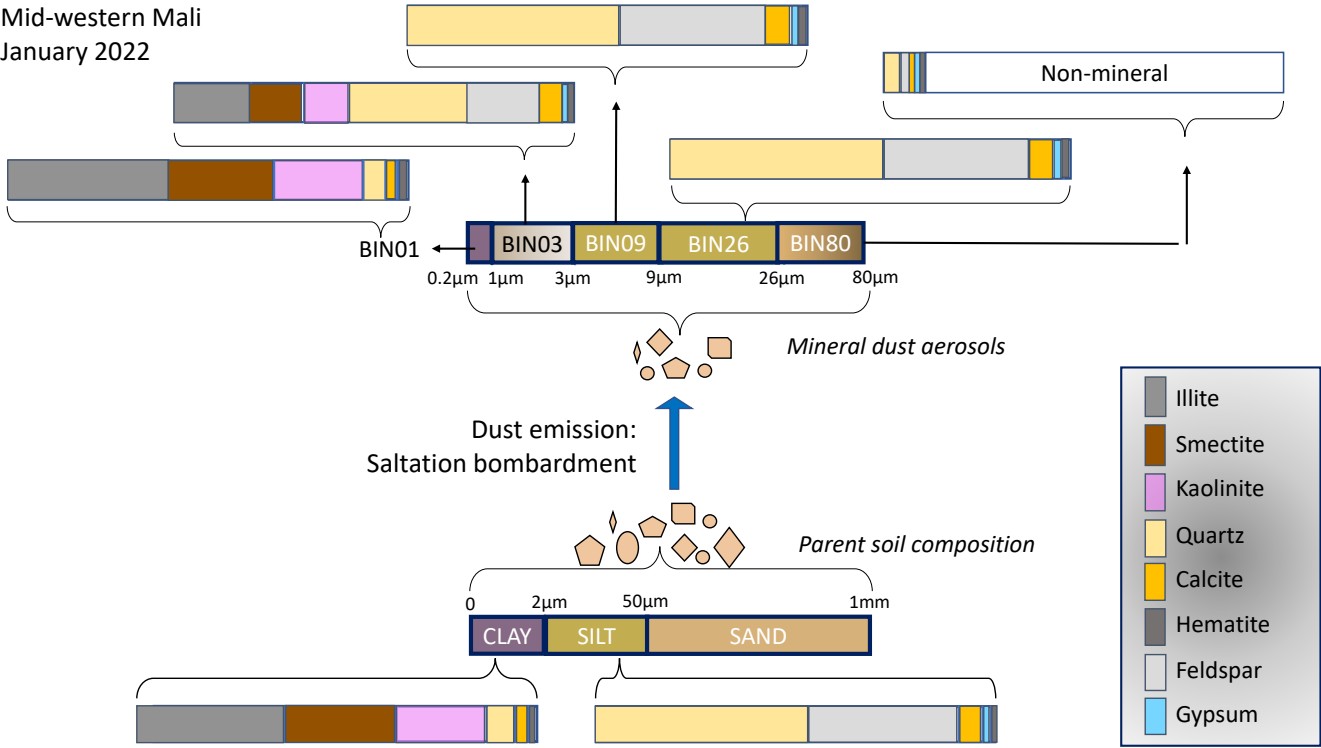

**Figure 1.** The scheme represents that mineral dust consists on a mixture of different minerals with different portions for both the mineral dust productive soil and aerosol size distributions. The BIN names are a MUSCAT convention that represents size classes in which the aerosol is transported, for more information see Table1. Boxes with a thicker line surrounding represent particle size distribution while the thinner line surrounding boxes represent mineral fractions. The minerals proportions depicted are representative for both mineral dust productive soil and aerosol compositions for a grid cell in Mid-western Mali (17.85°N, 4.85°W). Aerosol size distribution is taken for 27 January 2022 21:00 UTC.





Figure 2 represents the mineral fractions distribution for mineral dust aerosols in different regions. The mineral fractions are spread out for each of MUSCAT dust size classes according to their size distribution at the source soil. In Fig. 2, the relative mineral contribution to a normalized emission per size bin is shown. It is worth to mention that, for this approach, minerals are
considered as externally mixed, meaning that each particle of mineral dust is composed of an individual mineral, even though, in reality, mineral dust in the atmosphere is a mixture of internally and externally mixed minerals (Atkinson et al., 2013).

Figure 2 also illustrates the differences between the mineralogical composition on the Sahara Desert. In such a manner that the Saharan compositional fingerprints can be pointed out as indicated by Formenti et al. (2011, 2014) and Scheuvens et al. (2013). Specifically, a decrease in the illite/kaolinite ratio is observed from northern Africa (Algeria) towards the south western
Sahara Desert (Mali). An increment on calcite is also observed when moving southwards in the desert. Bodélé Depression is illustrated here for it being the most active mineral dust source in the world (Prospero et al., 2002) and a region with a distinct mineral composition; where there is a very low calcite content combined with a very low kaolinite to illite ratio due to a high kaolinite content.

## 2.4 Observational data for model evaluation

AERONET (AErosol RObotic NETwork) provides sun photometer measurements (Holben et al., 1998) which are used for a quantitative model evaluation. AOT is measured at 500 nm wavelength and at 675 nm from where the values for AOT at 550 nm are interpolated and compared with modelled values. Stations around the Sahara Desert are selected for the period of January-February 2022. The considered stations are: Mindelo, located at São Vicente (Cape Verde), Santa Cruz (Tenerife) off the Northwest African coast, Dakar Belair in Senegal, IER Cinzana in Mali and Banizoumbou in Niger. AERONET data
used were cloud-screened, calibrated and are AERONET indicated by the quality level of 2.0 for the stations: Mindelo, Dakar Belair, Santa Cruz Tenerife, and Banizoumbou, and where quality level 2.0 data were not available, cloud-screened data, level 1.5 data are used (IER Cinzana).

Furthermore, the spatial and temporal distribution of the modelled dust is evaluated by comparing the AOT derived from the modelled dust and AOT obtained by satellite retrievals. The satellite retrieved AOT used here is the Land Ocean Mean
AOT at 550 nm obtained through the Visible Infrared Imaging Radiometer Suite (VIIRS) instruments on board of the Suomi National Polar-Orbiting Partnership (S-NPP) and the Deep Blue Level 3 retrieval algorithm (Sayer et al., 2018). A regridding is necessary for comparison purposes in the sense that the model results are regridded to the coarser 1°x1° satellite retrieved data.

The evaluation of the model capacity to simulate mineral concentrations is done by comparing with former in-situ measure-
ments. Aerosol mineral specific mass concentrations measurements are sparse and rare. From the gathered in-situ mineralogical measurements, a selection was done based on specific meteorological conditions. COSMO-MUSCAT mineral data is, in this instance, restricted to January and February of 2022, and taking into account that mineral dust transport changes throughout the year (Schepanski et al., 2009), the meteorological conditions need to be taken into account for a more precise comparison. From the literature review, a first selection was done for the measurements inside the model domain. Then if the measurements



**Figure 2.** Relative contribution of each mineral emission flux to the BIN's total emission flux. The BIN names are a MUSCAT convention that represents size classes in which the aerosol is transported, for more information see Table1. Dust emission fluxes are calculated for 27 January 2022 21:00 UTC. Entrainment into the atmosphere is then conducted via these five size classes where each class is fractioned by their mineral composition as portrayed. Coordinates of the locations are as follows: Algeria (31.1°N, 1.9°E), Mauritania (21.25°N, 15°W), Mali (17.85°N, 4.9°W), Bodélé Depression (17°N, 18°E).

correspond to more than two dust events a general consideration was made regarding two classifications of meteorological conditions, one for the northern hemispheric summer months and another for the northern hemispheric winter months. Since the modelled period is part of the second classification, the reported mineral mass concentrations were compared for the mean of the whole modelled period. In the case where the measurements just represent one or two dust events, then the meteorological conditions for the specific case were investigated through both archive meteorological data and archive satellite retrievals,



specifically the Dust RGB product from the MSG-SEVIRI satellite archive (Schmetz et al., 2002). In this way a general idea of the dust source area is derived, and if a similar source is found on the modelled period then the reported mineral mass concentrations are compared with the simulated mineral mass concentrations from those individual dust events. The measurements that were selected following the criteria can be found in the Appendix A1

The evaluation of the model's capacity of reproducing the vertical mineral dust aerosol layering is also performed over 270 an specific location by comparing simulated mineral dust concentration with results from aerosol lidar measurements. Lidar measurements were taken with an automated multiwavelength Raman polarization and water-vapor lidar, Polly$^{XT}$ (Engelmann et al., 2016) located at the OSCM (Ocean Science Center Mindelo) in Mindelo, São Vicente, Cape Verde. From these measurements, optical properties that aid in the characterization of aerosols can be obtained by following a retrieval method based on a combined Raman and elastic-backscattered signals (Baars et al., 2016; Hofer et al., 2017; Haarig et al., 2022), where 275 both extensive, i.e., backscattering and extinction coefficients, and intensive, i.e., lidar ratio, Ångstrom exponent and depolarization ratio, optical properties can be retrieved. For Polly$^{XT}$ these retrievals can be done at three wavelenghts, at 355 nm, 532 nm and 1064 nm wavelengths. A detail explanation on how the dust mass fraction is derived from lidar retrievals is given in Section 2.5.3. The advantage of the comparison with lidar data is that it can indicate two things implicitly in a positive comparison: first, it can be used to confirm the simulated data, and second, some measured lidar data sets can hint to different 280 Saharan origins by linking the measurements to the mineral resolved emissions of dust.

## 2.5 Experimental setup

### 2.5.1 Model setup

The COSMO-MUSCAT model domain is set up to include the major part of the Sahara Desert and Cape Verde and constrained by the following coordinates: 30.75°W, 38.49°N – 39.32°E, 0.38°S. The horizontal grid spacing is 0.25° ( 28 km) and the 285 vertical resolution contains 40 levels, with a layer thickness of 20 m for the first (bottom) layer. The meteorological data, that act as initial and boundary data for COSMO-MUSCAT, is provide by the DWD with 3 hourly wind fields for the period January-February 2022. The model runs are reinitialized in overlapping cycles every 48 hours in order to keep the meteorology updated. For each 48 hours run, COSMO has a spin-up time of 24 hours, where MUSCAT is not running. After the spin-up time, MUSCAT starts running in parallel and computes the aerosol transport processes for the next 24 hours in order to complete the 290 cycle. Aerosols calculations are obtained from the second part of the described cycle. For the process to start again, COSMO is reinitialized 24 hours before the last MUSCAT cycle finished and when MUSCAT starts running again it uses the data produced on the last cycle as initial conditions.

The soil input files used for MUSCAT are: dust activation frequency map derived from MSG-SEVIRI IR-channels (Schepanski et al., 2007), soil vegetation file from Copernicus Global Land Service (Fuster et al., 2020), soil moisture file from the ERA5 295 land hourly data (Muñoz Sabater and Service, 2019), aeolian roughness length data set developed by Prigent et al. (2005), soil particle size distribution obtained from the SoilGrids database (Poggio et al., 2021), and the in this study newly introduced the





mineralogical database GMINER (Nickovic et al., 2012). The use of these soil data sets and the domain constraints have as a consequence that only continental mineral dust sources are regarded.

### 2.5.2 Model diagnostic

Aerosol optical thickness (AOT) is a measure of particle load in the atmosphere. Dust emission and deposition fluxes measurements are not extensively available or easy to realize (Schepanski et al., 2017), but the amount of mineral dust in the atmosphere can be assessed through their AOT. AOT measurements at 550 nm from both ground-based and space-borne remote sensing instruments are extensively available. Hence, AOT is chosen for evaluating the model's ability of simulating mineral dust life cycle. Noteworthy for the comparison is that the model considers only mineral dust aerosols, whereas, the retrieved AOT is

done via remote sensing measurements that are affected by all types of atmospheric aerosol particles. However, AOT measurements for the studied region are considered to be dominated by mineral dust. Validating the simulated mineral dust aerosol loading through the comparison with AOT retrievals is quite common for the Sahara Desert as in Heinold et al. (2011, 2016); Schepanski et al. (2015, 2016, 2017) and Tegen et al. (2013).

AOTs for the 550 nm wavelength from the simulated mineral dust aerosol concentrations are calculated following:

$$
\tau = \frac{3}{4} \sum_j \frac{Q_{ext,550nm}(r_{eff}(j))}{r_{eff}(j)\rho_{dust}} M(j),
\tag{4}
$$

where $Q_{ext,550nm}$ is the dimensionless dust extinction efficiency at 550 nm that varies according with the particle size class $j$ effective radius $r_{eff}(j)$, $\rho_{dust}$ is the particle density, set at 2650 kg/m$^3$ and $M(j)$ is the column mass load of mineral dust given in kg/m$^2$, obtained from the simulation results. The values of $Q_{ext,550nm}$ (Table 1) are calculated from refractive indices computed by Sokolik and Toon (1996) which consider for their calculation a fixed composition for dust; although it

is documented that variations in dust composition affect the particles optical properties (Journet et al., 2014; Wagner et al., 2012). Nonetheless, this calculation is used as a first approximation for comparison purposes with the retrieved AOT obtained from both the ground-based sun-photometers from the AERONET (Holben et al., 1998), and from the satellite instrument, the Visible Infrared Imaging Suite (VIIRS).

### 2.5.3 Lidar based dust mass fraction

Polly$^{XT}$ lidar measurements include the retrieval of aerosols optical properties. Such measurements are influenced by other aerosols besides mineral dust. Tesche et al. (2011b) show how to separate the optical properties of dust and non-dust components. An application of such separation is implemented for this study, where we aim at a direct comparison by calculating mineral dust mass fractions from the Polly$^{XT}$ retrieved optical properties. A one to one comparison can then be made with the simulated mineral dust mass concentrations at the grid cell where Mindelo is included. For the calculation of the mineral dust

mass fraction from the lidar measurements, the method POLIPHON presented in Ansmann et al. (2019) is used. The method uses conversion parameters determined from AERONET aerosol climatologies that divide the aerosol load between mineral dust aerosols and non-dust aerosols. Specifically for the calculation of mineral dust and non-dust mass fractions, the particle depolarization ratios at 532 nm are used to calculate the fractions of mineral dust and non-dust aerosols in the backscatter and





extinction coefficients. Backscatter coefficients at 532 nm are then used for calculating the final dust mass fraction. Several as-
sumptions are included in the POLIPHON method, for instance, the homogenization of optical properties values for all mineral
dust particles disregard the dust source region, and therefore disregarding compositional differences.

For the single point measurement comparison, it is important to consider that local measurements represent values at that
single point while the model provides values for the whole grid cell.

## 3   Results

This section contains the results for the specific model setup based on the Sahara Desert including the Cape Verde archipelago
as detailed in Section 2.5.1. The model, COSMO-MUSCAT, simulates the mineral dust life cycle for the months of January-
February 2022 over the region, and now includes mineralogical information. The section starts with showing comparisons
of the atmospheric aerosol loading through AOT calculations as explained in Section 2.5.2. AERONET and VIIRS/SNPP
AOT values are compared with the model derived AOT values. Afterwards, the ability of the model to simulating mineral
mass content is shown via comparisons with field measurements and by displaying the mineral distribution throughout the
domain via total column mineral mass maps. Lastly, local comparisons are done for São Vicente, Cape Verde. We compare the
simulated mineral dust mass concentration vertical profile for the grid cell where São Vicente is with the lidar derived mineral
dust mass concentration vertical profile. The lidar derived dust mass concentration is obtained from the lidar Polly$^{XT}$ retrieved
optical properties as described in Section 2.5.3. In addition a temporal evolution of column dust mass concentrations from the
simulating results and from the lidar retrievals are compared for the time period of 1 Feb to 2 Feb 2022 early morning.

### 3.1   Evaluation of modelled dust aerosol optical thickness

### 3.1.1   COSMO-MUSCAT vs AERONET comparison

Figure 3 shows the comparison between simulation based AOT calculations and the AOT from AERONET. The selected
AERONET stations are directly affected by the Saharan dust plumes for the studied period: January-February 2022. The
locations of these stations are shown in Fig. 3f. The stations were chosen downwind of the source regions and on the dust
transport pathways towards the Atlantic, specifically towards Cape Verde.

The simulated dust AOT magnitudes and temporal variabilites agree very well with AERONET results (Fig. 3a-e). A single
exception is found for the station Santa Cruz Tenerife on 29 January 2022, where the AOT peak was much higher than the
simulated values. That day an anticyclone system formed over the region that allowed for a heavy dust load to be swiftly
transported from central Sahara towards Tenerife. COSMO replicates that situation well but shifts the center of the pressure
system for a couple of degrees northward, which changed the simulated dust transport route. The shifting of the pressure system
brought two consequences that caused COSMO-MUSCAT's low mineral dust AOT for Tenerife on 29 January 2022: (1) Most
of the simulated dust plume was transported north of Tenerife and therefore is not captured on the grid cell that corresponds
to the sun photometer's location. (2) Due to the change on the pressure system position, precipitation that occured in the area



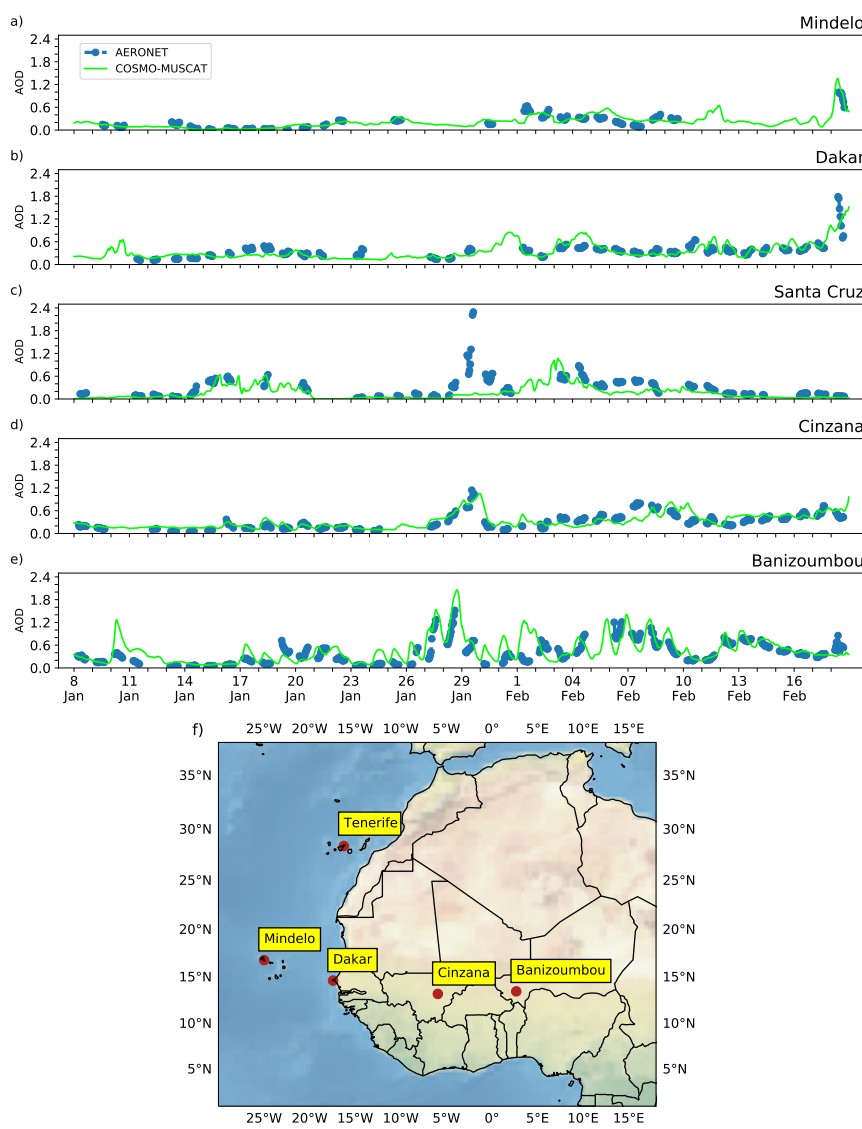

**Figure 3.** Dust AOT at 550 nm calculated from COSMO-MUSCAT dust concentration fields (green) and AOT at 550 nm from AERONET sun-photometer measurements (blue) for January-February 2022 where each x-axis tick represents the 12:00 UTC for each day in the range of 8 January to 18 February 2022. Five different stations across the Sahara Desert and downwind locations are shown. a) Mindelo (16.878°N, 24.995°W; Cape Verde; Level2.0), b) Dakar Belair (14.702°N, 17.426°W; Senegal; Level2.0), c) Santa Cruz Tenerife (28.473°N, 16.247°W; Spain; Level2.0), d) IER Cinzana (13.278°N, 5.934°W; Mali; Level1.5), e) Banizoumbou (13.547°N, 2.665°E; Niger; Level2.0), (f) AERONET stations geographic locations and names used in this comparison.





moved to Tenerife on 29 January 2022, which activated MUSCAT's mineral dust wet deposition scheme resulting in even less mineral dust remaining in the atmosphere.

Noteworthy is the comparison with the station at Cinzana since the compared values are always on the same range of magnitude, with the biggest AOT difference being 0.5, and the temporal evolution is well portrayed by the modelled values. Moreover it needs to be kept in mind that the model derived AOT is due to mineral dust only while the AERONET measured
AOT is influenced by a wider array of aerosol types, and therefore, instances where the AERONET AOT values are bigger than the simulated derived AOTs are expected.

### 3.1.2   COSMO-MUSCAT vs VIIRS/SNPP comparison

The spatial and temporal distribution of atmospheric mineral dust is illustrated for the days 31 January to 2 February 2022 in Fig. 4. Maps of model derived mineral dust AOT at 550 nm, the VIIRS/SNPP AOT retrievals, at the same wavelength, and
the difference (COSMO-MUSCAT AOT minus VIIRS/SNPP AOT) are shown. Gray areas in the VIIRS/SNPP maps indicate cloudiness that obscured the aerosol plume and therefore no AOT retrievals were possible. A strong mineral dust outbreak

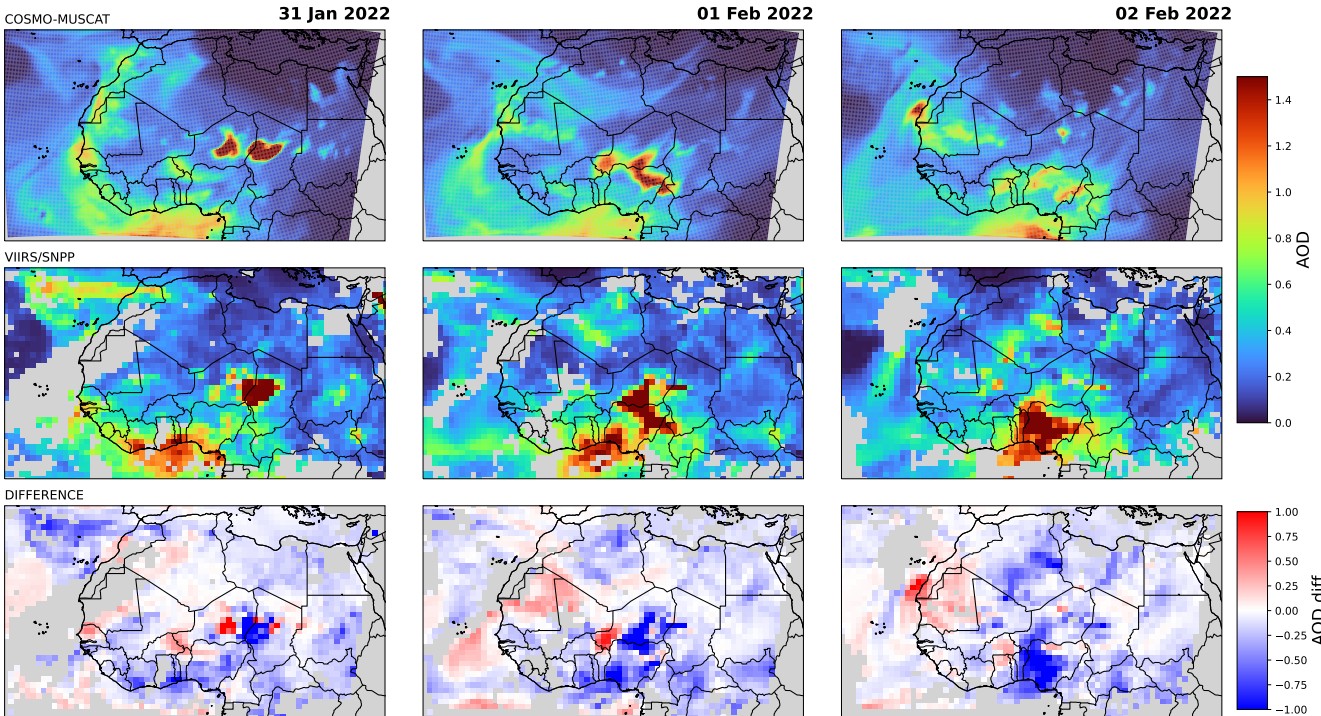

**Figure 4.** The evolution of two dust outbreaks in the Sahara Desert for 31 January, 1 February and 2 February 2022. Maps are shown for 12:00 UTC. Color shading represent AOT at 550 nm (top and middle row). Gray shading indicates no data. First row shows maps of COSMO-MUSCAT derived dust AOT. Second row shows the maps of AOT retrieved by VIIRS/SNPP. Third row shows the difference between model and observations, that being, model derived AOT – satellite retrieved AOT.





across the Sahara leading towards the Atlantic Ocean ocurred during these days. The model simulates active dust source regions across West Africa and the Bodélé Depression. Simulated dust plumes above Niger and around the Bodélé Depression in northern Chad match temporally well with VIIRS/SNPP observations, whereas differences are present for dust source regions

across the western parts of the Sahara Desert (e.g., Mauritania). The modelled dust AOT above Senegal, Mauritania, West Sahara, and Morocco cannot be fully compared with VIIRS/SNPP retrievals due to cloud coverage. This is evident e.g., on 31 January in the western Sahara emissions.

Over Mali and Mauritania, simulated AOTs are slighlty overstimated compared to the MODIS retrievals. In contrast, AOTs are underestimated over the Bodélé Depression, which is known as mineral dust emission hotspot. For example, on the 31

January both VIIRS/SNPP and COSMO-MUSCAT show high AOTs over the Bodélé Depression and Niger, but the difference map shows that the model is over predicting AOTs over Niger and under predicting above the Bodélé Depression. On average and for the period considered, the difference maps shows that the simulation based AOT values agree reasonably well to the VIIRS/SNPP retrieved AOT.

For 1 February, VIIRS/SNPP shows high AOT values both on the southern and northern parts of Nigeria. The northern part

matches COSMO-MUSCAT derived AOT and can be therefore attributed to mineral dust aerosols, although from the difference map, COSMO-MUSCAT seems to be underestimating the amount of dust in the atmosphere. High dust emissions are shown by the simulations for the day before (emissions not shown on the figure) over the southern part of Niger. While the southern Nigeria high AOT spot shown in VIIRS/SNPP does not appear in the simulated results. However the area is not recognized as a highly active mineral dust source region but it is indeed related to high aerosol emissions due to biomass burning during the

northern hemispheric winter (Heinold et al., 2011; Tesche et al., 2011b). Such aerosols are most likely the reason for a higher AOT over the region. A similar behavior is found as well on 2 February above the same region, where the highest differences of the whole domain are found.

Dust plume pathways can be traced both in the model and in VIIRS/SNPP maps through the three days. For 31 January, mineral dust is located over the Bodélé Depression and it travels southwards in direction towards the equatorial Atlantic Ocean

as seen in both plots for 1 and 2 February. At the same time another dust plume is observed over Mauritania and travelling over the Atlantic Ocean, heading towards Cape Verde. Noteworthy is that the differences above the Cape Verde region are almost zero, pointing to a very good agreement between VIIRS/SNPP and the simulated results. The pathways of the mineral dust plumes cannot be fully compared since there is no more VIIRS/SNPP information over the Atlantic towards Cape Verde for the 31 January and 1 February and over the equatorial Atlantic Ocean on the 2 February. A negative bias of around 10 % over

bright surfaces for VIIRS/SNPP AOT retrievals has been reported when compared to AERONET values (Sayer et al., 2019). A tendency towards more negative biases in the bright regime is also reported when AOT values grow bigger than 1.0 (Sayer et al., 2019). This negative bias affects the comparison of model and VIIRS/SNPP AOT.

## 3.2 Evaluation of modelled mineral dust composition

Measurements of mineral dust aerosol composition are rare and sparse across the Sahara Desert, nevertheless, some have been

reported throughout the years. For the comparison of modeled mineral composition with measurements shown in Fig. 5, we





used a specific data set of past measurements from that matched the best our simulated meteorological conditions as explained in Section 2.4.

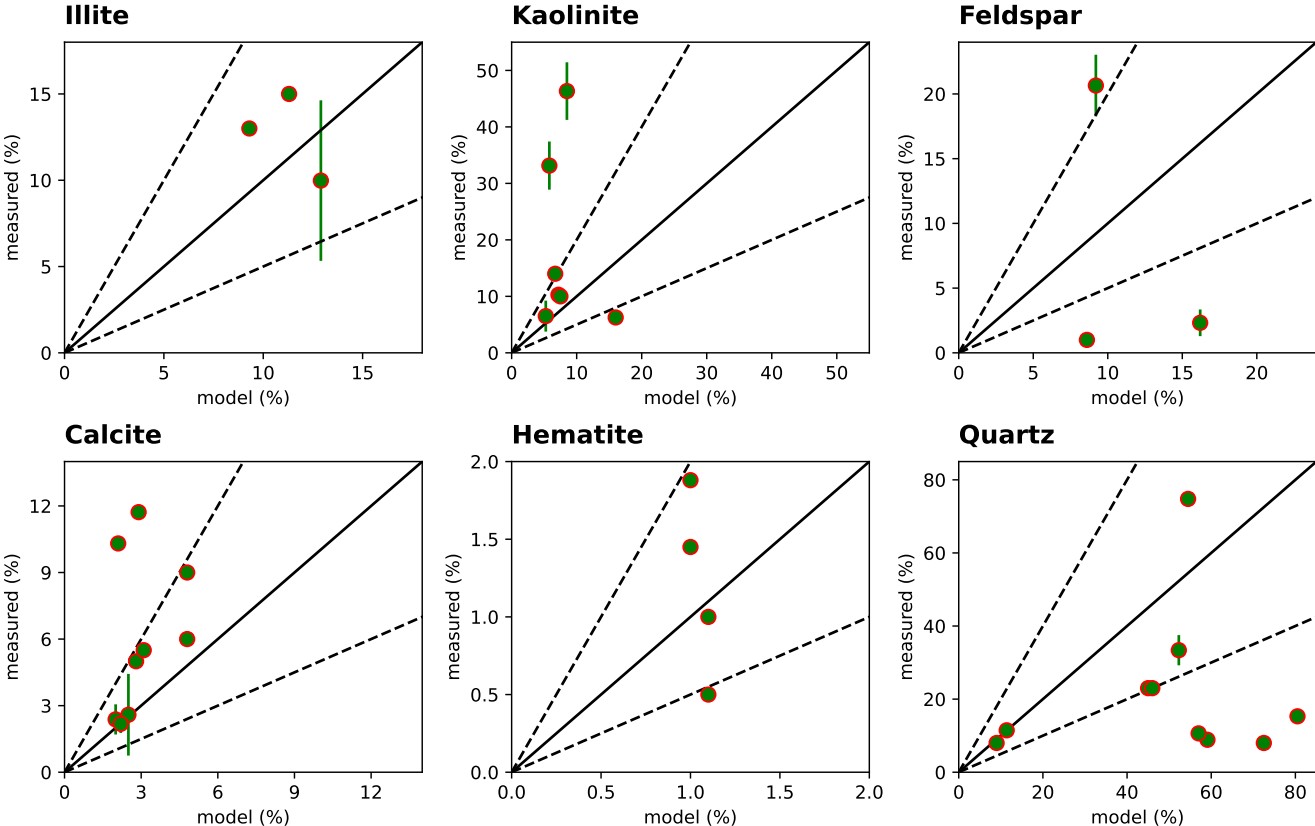

**Figure 5.** Scatterplot of minerals mass percentages of illite, kaolinite, feldspar, calcite, hematite and quartz simulated by COSMO-MUSCAT vs. measurements. All measurements were done on aerosols bulk size range. The dashed lines represent the ratios of 2:1 and 1:2 between the simulated and observed mineral percentages. The error bars are present when reported in the measurements.

Figure 5 shows two rows of individual mineral comparisons of modelled vs. measured mineral data. Both are represented as a percentage of the dust aerosol mass. The phyllosilicate minerals (i.e., illite and kaolinite) shown on the figure are on average underrepresented by the model. Feldspar and quartz minerals are special cases since different definitions of these minerals are used for different set of measurements (Formenti et al., 2008; Jeong and Achterberg, 2014) while the mineral fractions described in the GMINER database contain a broad definition of these two minerals. Hematite comparisons show good agreement with measurements, nevertheless the model calculates similar amounts of mineral mass for all the dust events, which is not representative for the diversity of this mineral on source regions. Calcite comparisons are noteworthy since they fit best within the whole mineral comparison array. Calcite and quartz are the only minerals in the model that are represented





in both clay and silt sizes. As seen in Perlwitz et al. (2015a) better agreements between modelled and measured mineral data was found when considering a reaggregation coefficient that appoints mineral fractions to both clay and silt sizes.

**Figure 6.** COSMO-MUSCAT total column mineral dust mass concentration [mg/m$^2$] for 2 February 2022 at 05:00 UTC of minerals mass concentration in mg/m$^2$. a) Illite concentration, b) kaolinite concentration, c) hematite concentration, d) calcite concentration. Note the logarithmic scale.

The atmospheric column mass concentrations for individual minerals over the whole region from COSMO-MUSCAT are shown in Fig. 6 for 2 February 2022 at 05:00 UTC. The simulated atmospheric mineral mass column that mimics mineral composition at the mineral dust productive soil. As a consequence, we observe high kaolinite values that came from dust emissions over central Mali, which has a higher content of kaolinite than the other activate dust emission regions on this time (see Fig. 2 for more information on the mineral fractions of this specific location). High illite aerosol concentrations are found both on the south-eastern Sahara as well as in the northwestern Sahara, resembling illite concentrations on the mineral dust productive soil as can be seen in Formenti et al. (2011, 2014); Nickovic et al. (2012) and Scheuvens et al. (2013). Illite soil concentrations generally increase when moving towards the north and west of the Sahara Desert. Hematite concentrations are





rather low in the whole domain with some exceptions above Nigeria and Cameroon. Most likely this higher hematite aerosol concentration is a consequence of mineral dust emitted from Niger and transported southwards, since the higher soil hematite content for the whole desert are found in south Niger and central Mali. Calcite content is high east of the Atlas Mountains, where the calcite soil concentrations are the highest for the whole region. Rising calcite concentrations over the southward-eastern Sahara can also be seen in Fig. 6, which may be due to some significant calcite soil concentrations found where kaolinite soil content is the highest (Formenti et al., 2014; Nickovic et al., 2012). The high illite content over the Atlantic Ocean comes from the western Sahara dust emissions as pointed out in Section 3.1.2. Interestingly, concentrations of both kaolinite and illite over Cape Verde show similar mass loadings, which points to the differences on mineral fractions from the two different dust active regions seen in Fig. 4. The western Sahara dust emitting region is known as having a high illite to kaolinite ratio, while the Sahelian dust emitting area is known for containing a larger amount of kaolinite. The column concentrations of the minerals at Cape Verde are due to mixing of these two plumes.

Hematite concentrations are shown in Fig. 6 because of its distinct optical properties. Most of the minerals found in the GMINER data set have similar optical properties in the shortwave part of the spectrum, from 200 nm to 4000 nm (Journet et al., 2014). The exception are the minerals that are considered as proxies for iron oxide content, that being, hematite and goethite (Lafon et al., 2004; Wagner et al., 2012; Zhang et al., 2001). In the GMINER data set, the hematite mineral represents both the hematite and the goethite mineral content. Hematite and goethite are more absorbing, in the UV part of the spectrum (Wagner et al., 2012; Di Biagio et al., 2019), than the other minerals found in mineral dust. Noteworthy is that even amounts as low as 1.4 to 2.5 % hematite content in volume can significantly modify mineral dust direct radiative effect. Balkanski et al. (2007) show that a change of 0.6 % of hematite content, in weighted volume, has an impact of 4 W/m$^2$ column heating. Therefore, the mass concentration of a certain mineral in the atmosphere does not directly translate to its radiative impact. Relatively small quantities of hematite content in mineral dust aerosol dominates the shortwave interaction and has direct effects to atmospheric column heating.

## 3.3 Comparison to lidar remote sensing and model-based attribution of loca mineralogical properties

Mineral dust is mixed up in the atmosphere over the African continent by turbulent mixing within the planetary boundary layer reaching up to 3–5 km and gets subsequently transported following the regional meteorological drivers. During such transport, the mineral dust plume can remain near the surface or can be transported in elevated layers over the adjacent Atlantic Ocean. The elevation of the Saharan dust layer depends on the season; during northern hemispheric winter, transported Saharan mineral dust is mostly observed at near-surface layers (Schepanski et al., 2009).

From 1 February 18:00 UTC to 2 February 2022 06:00 UTC, two lofted aerosol layers were observed by the lidar Polly$^{XT}$ above Mindelo, São Vicente. To determine whether and how much dust is present in these two lofted aerosol layers, a fine temporal resolution is needed to avoid averaging over dust and non-dust parts of the retrieved profiles. From the finer temporal resolution, optical properties can be retrieved, which can confirm whether the measured values fit the mineral dust optical properties reference values (see Table 2 in Tesche et al. (2011a)). For this particular day, a specific time period between 04:30 and 05:29 UTC for 2 February 2022 was selected as the finer temporal resolution. Figure 7a shows two of the optical properties





retrieved for the 532 nm wavelength channel, where the backscattering coefficient ($\beta$) is shown in light green and the particle depolarization ratio ($\delta$) in dark blue.

In Fig. 7, the two lofted aerosols layers are clearly visible; the first one starts at 650 m and ends at 1.4 km while the second layer starts at 1.6 km and ends at around 4.5 km. In order to appoint the aerosol types found in the layers, an analysis of the lidar-retrieved intensive (i.e. independent of the aerosol amount) optical properties is necessary. From this retrieval, both lidar

ratios (LR) at 355 nm and 532 nm wavelengths are, for the first lofted aerosol layer: 65 sr and 60 sr, and for the second lofted aerosol layer: 60 sr and 57 sr, numbers which fit the range specified by Tesche et al. (2011a) characterizing smoke and mineral dust mixture, (i.e., 67±14 sr and 67±12 sr). Furthermore, the particle depolarization ratios ($\delta$) for both wavelengths are for the lower lofted layer: 0.15 and 0.17, and for the upper lofted layer: 0.17 and 0.23, values which are too low to point to pure mineral dust in the layers, and rather again correspond well with the range that Tesche et al. (2011a) identified as smoke

and dust mixture (i.e., 0.16± 0.04 and 0.16± 0.03). Therefore, the intensive aerosol optical properties retrieved from Polly$^{XT}$ signals from 2 February 2022 04:30–05:29 UTC reveal that both lofted aerosol layers are composed of a mixture of smoke and mineral dust.

Figure 7 also shows the results of applying the POLIPHON method (see Section 2.5.3) to the vertical profiles of optical properties measured with lidar. The POLIPHON result shows clearly that a mixture of dust with non-dust aerosol particles is

present in the atmosphere above Mindelo (Fig. 7a,b). The simulated vertical profile of mineral dust from COSMO-MUSCAT is pictured in Fig. 7c. The total mineral dust mass concentration is shown by the orange line, while the other lines represent the mass concentrations of: quartz, illite, kaolinite, calcite and hematite. The simulated profile does not represent the layered structure of the vertical aerosol distribution. The lidar retrieved vertical profile on the other hand shows two separated lofted dust layers due to an aerosol free air mass intrusion with a 100 m thickness between them. The thickness of the aerosol free layer

that separates the two layers is thinner than the correspondent vertical level at this altitude in the model, which extends from 1229–1466 m. The model vertical resolution is therefore too coarse to be able to simulate the clean air mass intrusion in between the lofted mineral dust layers. This fact has as a consequence that the simulated dust layer appears to be one dust layer in the model instead of the layered mineral dust plumes observed by the lidar. The differences between the lidar retrieved vertical profile and the simulated profile include the two lofted layer structure shown on the lidar results, and the specific heights at

which the maximum of mineral dust mass concentrations are found. From the Polly$^{XT}$ derived dust mass concentration vertical profile, the peak of dust mass concentration for the lower lofted layer is found at 1.1 km with a mineral dust mass concentration of 101 µg/m$^3$, while for the upper lofted layer, the peak is found at 2 km with a mineral dust mass concentration of 156 µg/m$^3$. The simulated profile shows one single dust layer with elevated dust mass concentrations between 0.5 km and 3.5 km and a peak dust mass concentration of 136 µg/m$^3$ at 1.5 km height. COSMO-MUSCAT positions the lofted dust layer peak 500 m

meters below from the peak of the upper lofted layer retrieved from the lidar signals, but the peak dust mass concentrations differ by 20 µg/m$^3$. For the lower lofted layer, the dust mass concentration peak obtained from POLIPHON compares well with the dust mass concentration calculated by COSMO-MUSCAT at the same height, since the model calculates 115 µg/m$^3$ for 1.1 km.

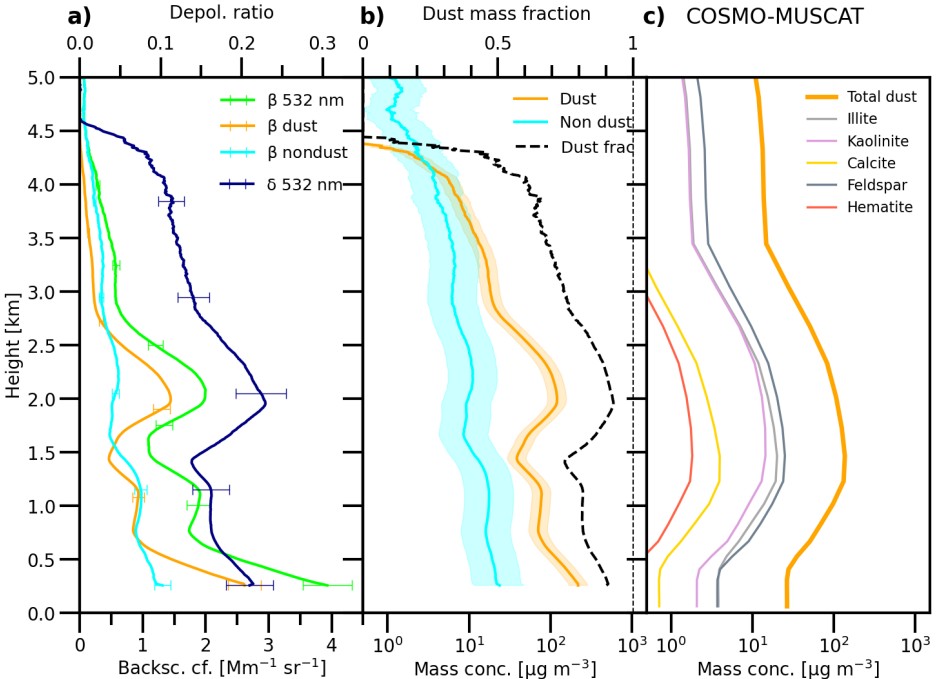

**Figure 7.** Vertical profiles of lidar retrieved optical and microphysical properties (obtained by applying the POLIPHON method to the polarization lidar measurements) and modelled mineralogical composition. The lidar measurement was performed on 2 February 2022 04:30–05:29 UTC in Mindelo. The 532 nm particle backscatter coefficient (a, light green) and the particle linear depolarization ratio (a, dark blue) are the input to obtain the separated dust and non-dust profiles in (a,b). The POLIPHON products are the derived 532 nm dust backscatter coefficient (a, orange) and the non-dust backscatter coefficient (a, light blue), dust mass concentration (b, orange) and the non-dust mass concentration (b, light blue), and the dust mass fraction (b, black, ratio of the dust to total particle mass concentration, dashed black vertical line indicates a dust mass fraction of 1). Vertical profiles of simulated mineral dust mass concentrations from the COSMO-MUSCAT model (c). The vertical profile corresponds to values calculated for the grid cell where Mindelo, Cape Verde is found in the model. Each dot represents the concentration at the middle of the vertical model layer. Total mineral dust mass concentration (c, orange) is shown together with the mass contributions of some selected minerals.

The dust mass concentration differences below the lower lofted layer need a more detailed discussion. From the model data, the mineral dust above Mindelo for 2 February at 05:00 UTC came from two different source regions in the Sahara as shown in Fig. 8a. The modelled air masses that transported dust towards the Cape Verde on the lowest altitudes came from northwestern Sahara (back trajectory (a) in Fig. 8a). Along the trajectory of these air masses, where dust was in transit in between the Canary islands and the Cape Verde archipelago, clouds were prominent, as depicted for 31 January in Fig. 4. From satellite retrievals (not shown) precipitation took place in the vicinity the pathway of these air masses transporting dust at near surface altitudes. COSMO-MUSCAT simulates the dust transport and precipitation, but the precipitation area simulated is larger than the one shown by satellite retrievals. The area covered by the simulated precipitation affected directly the modelled dust transport



pathway. Precipitation activated the wet deposition scheme and mineral dust was lost in the simulated transport. Furthermore, it cannot be discarded that some of the mineral dust captured by the lidar signals below 650 m altitude could be due to local mineral dust emissions, which are not included in the model. Nonetheless, it should be kept in mind that for single-point

measurements and modelling results comparisons, local measurements may not be entirely representative of a whole model grid cell (28x28 km$^2$).

Figure 7c illustrates specific mineral mass concentrations, where quartz represents the highest contribution to the total simulated mineral dust. Illite and kaolinite have similar contributions, but their vertical distribution differ due to the different dust source regions. The different origins depicted by Fig. 8a differ in their illite to kaolinite content, where, the Northwest

African source region has a higher illite content, and the source regions corresponding to the Sahel area have a higher kaolinite content. Consequently, the vertical profile simulated above São Vicente shows a higher amount of illite mass concentration at lower altitudes whereas the kaolinite mass concentration increases at a higher altitude.

Hematite is here included as it is assumed to represent the iron oxides content, which are known to absorb in the UV wavelength and therefore influence optical retrievals (Wagner et al., 2012; Di Biagio et al., 2019). Specifically, higher hematite

presence is linked to higher UV absorption which could be identified on the lidar retrieved intensive optical property, i.e., the lidar ratio, since this optical property is influenced only by shape and composition (Veselovskii et al., 2020). Hematite content on this day was almost negligible and consequently no significant UV signature can be observed from the lidar retrieved optical properties.

Figure 9 shows the comparison of the Polly$^{XT}$ derived and the simulated dust mass column concentrations above the planetary

boundary layer. The atmospheric column for this comparison starts above the planetary boundary layer for two reasons: (1) the lack of local sources in the model, and (2) depolarization measurements cannot be retrieved close to the surface due to Polly$^{XT}$ characteristics, therefore, no dust mass fraction can be derived.

In the Fig. 9, the Polly$^{XT}$ derived column dust mass concentrations vary more than the simulated column dust mass concentrations from 1 February 18:00 UTC to 2 February 06:00 UTC. For the first half of the time period considered here (i.e., 1

February 18:00 UTC to 2 February 01:00 UTC), the height of the planetary boundary layer is below the residual dust fraction found below the first lofted dust layer, and therefore Polly$^{XT}$ retrieval is more affected by it than on the second half of the comparison. On the model calculations, the residual dust fraction is under predicted as a consequence to the above mentioned meteorological and simulation conditions. For the second half of the time period considered (i.e., 2 February 01:00 UTC to 06:00 UTC), the planetary boundary layer is above the residual dust fraction and therefore lidar column dust mass concentra-

tions drop since it is only taking into account the lofted aerosol layers. For this half of the temporal comparison, the simulated column dust mass concentrations are very close to the the Polly$^{XT}$ retrieved column dust mass concentrations, which indicates that the lofted layers are well simulated.



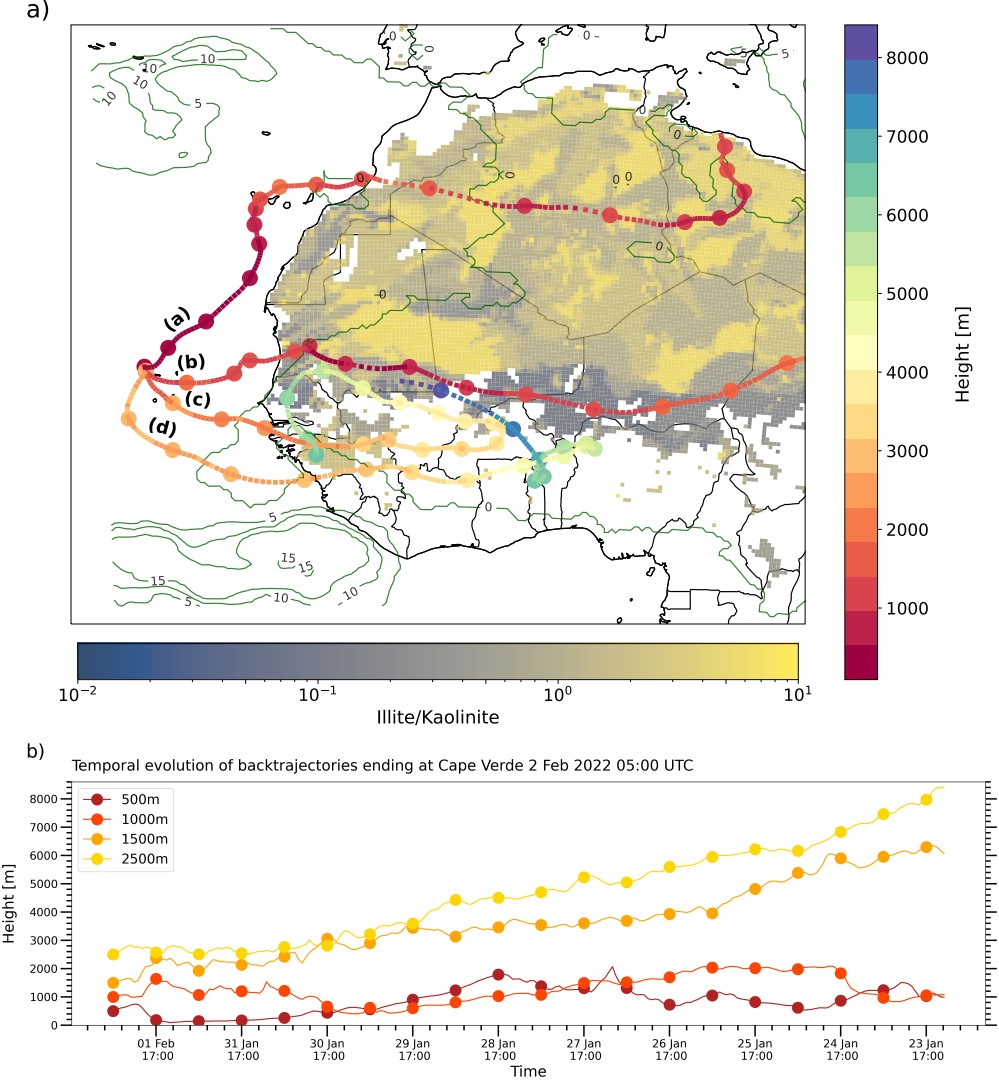

**Figure 8.** Backtrajectories calculated from COSMO-MUSCAT outputs using LAGRANTO (Miltenberger et al., 2013). a) shows backtrajectories starting on 2 February 05:00 UTC above São Vicente, Cape Verde, at the following heights above sea level: (a) 500 m, (b) 1000 m, (c) 1500 m and (d) 2500 m. Average precipitation from 30 January 00:00 UTC to 2 February 5:00 UTC is portrayed in contour lines. The illite to kaolinite ratio is shown on the background in blue and yellow colors, where blue shows the soils where kaolinite is predominant and yellow shows the soils where illite is the leading mineral fraction. b) shows the temporal evolution of the individual backtrajectories.

## 4 Discussion

The AOT and lidar comparisons are discussed first. Noteworthy is that for the AOT comparisons they are not intended to

yield the same results since both AERONET and VIIRS/SNPP estimate include all the aerosols that interact with radiation at



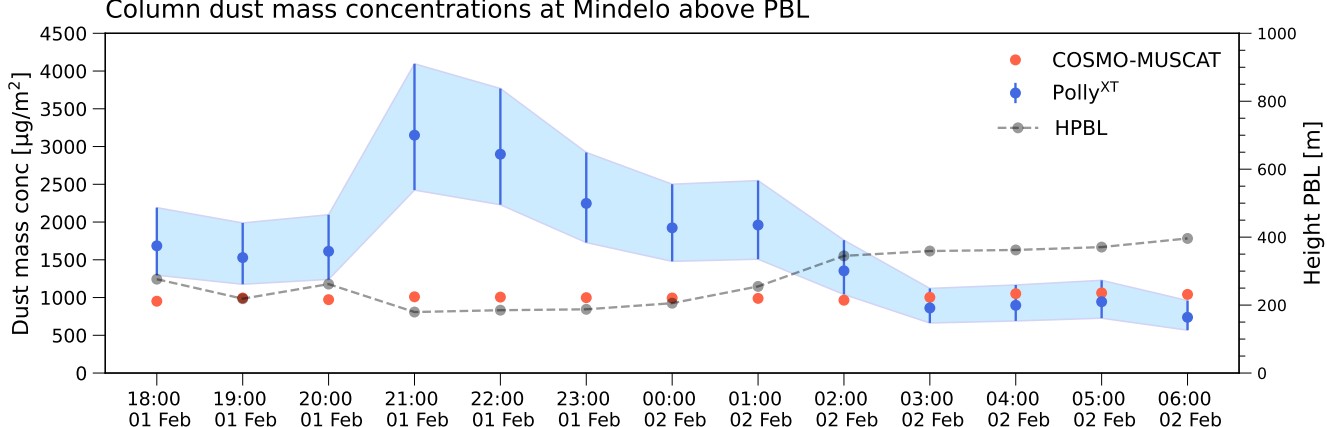

**Figure 9.** Column dust mass concentrations above the height of the planetary boundary layer. Temporal evolution of both PollyXT derived dust mass concentrations and COSMO-MUSCAT results for each hour between 1 February 2023 18:00 UTC and 2 February 2023 06:00 UTC. The height of the planetary boundary layer (HPBL) calculated from COSMO-MUSCAT is given by the black circles.

nm, while the model is only considering mineral dust aerosols. Nonetheless, the region is known for AOT retrievals which are dominated by mineral dust aerosols.

Compared to AERONET AOT (Fig.3), the model follows the temporal-spatial evolution very well and observed and modelled dust AOT agree. The same trend is found for the VIIRS/SNPP comparison (Fig.4), where the averages of the difference

maps are slightly negative, indicating somewhat higher values from the VIIRS/SNPP retrievals. Additionally, during northern hemispheric winter season the Sahel and the Gulf of Guinea regions are actively emitting aerosols due to biomass burning (Heinold et al., 2011; Tesche et al., 2011b), so that during this time of the year it is likely that the atmospheric burden due to aerosols will not be only due to mineral dust aerosols. Therefore, higher AOT values from VIIRS/SNPP than from the simulated results are expected since the model only simulates dust aerosols. Yet a definitive assessment regarding whether the simulated

mineral dust is under- or over predicted is difficult to obtain from this comparison, in part because of VIIRS/SNPP are reported to show negative biases over this region (Sayer et al., 2019). A negative bias could explain some differences for certain mineral dust hotspots, as Niger on 31 January, but does not explain the differences on the Bodéle Depression. In order to understand if the model is under representing Bodéle Depression dust emission more studies are required, and since the region is known for having very specific soil characteristics and requiring special considerations for its simulation (Tegen et al., 2006).

Generally, the AOT spatial distribution retrieved by VIIRS/SNPP that could be attributed in its majority to mineral dust, is similar to the calculated one from COSMO-MUSCAT simulations which suggests that the atmospheric dust life cycle including dust plumes trajectories on the studied period are well calculated. Both AERONET and VIIRS/SNPP AOT comparisons show a similar trend in the spatial-temporal coverage, as well as similar magnitudes as those measured. It can be concluded that overall COSMO-MUSCAT simulates the emission, transport and deposition realistically by including its relevant and determining

atmospheric processes.



The vertical distribution for 2 February 2022 at 5:00 UTC shown in Fig. 7 is reasonably well represented by the model. The two lofted layer structure is not shown on the modelling results. The air mass intrusion that separates the two mineral dust layers is too thin to be simulated, since the thickness of the vertical layer where this occurs is 237 m and the thickness of the air mass is roughly 100 m. Consequently, the two lofted layer structure cannot be simulated with the used vertical resolution.

A more significant difference is that the high dust mass concentrations below the lower lofted layer are not represented in the model. The model does not simulate this due to a simulated pressure system movement that shifted precipitation towards the trajectory of the dust plume over the Atlantic Ocean and due to the model not simulating any local emission for the Cape Verde archipelago.

From the comparison of the column dust mass concentrations it can be concluded that the model represents reasonably well
the vertical-temporal distribution of mineral dust layer. Nevertheless, the ability of the model to represent the vertical structure of mineral dust mass concentration in the atmosphere cannot be assessed based on one day example. Additionally, single-point measurements may not be entirely representative of a whole model grid cell, and further validation with lidar retrieved profiles are encouraged including validations among different seasons.

The above mentioned results depend on comparisons between the simulated mineral dust and measurements based on the
assumption of homogeneous mineral dust optical properties (i.e., both the AOT calculation from mineral dust mass concentrations and the POLIPHON method). The differences found on the simulated mineral dust and measurements discussed here could be in part attributed to this assumption. Balkanski et al. (2007) suggested that the reason for model over estimations of mineral dust lay on the discrepancy on mineral shortwave refractive indices. Their study shows, how, by modifying the homogeneously assumed optical properties of mineral dust, better agreements with AERONET measurements can be found.

The specific mineral mass composition comparisons illustrate a strong demand for more mineral composition measurements. The feldspar and quartz comparisons cannot be analyzed since different definitions of the minerals are used through the measurements. The results of illite and kaolinite comparisons can support the suggestion by Perlwitz et al. (2015b) that illite and kaolinite should also be considered on the silt size regime, since the comparisons show that, on average, they are under-represented in the model. Calcite that is considered in both silt and clay size ranges in the model compare best to the
observations. Hematite composition compares well to measurements, but due to the scarcity of the measurements, no further assessment can be made. Nonetheless, is it noteworthy that the most of the measurements fall in the same order of magnitude as the simulated results.

The results suggest that the model is successfully including mineralogy inside the simulation of the dust life cycle. The results also show that the model is able to reproduce the meteorological drivers that lead to a specific seasonal behavior.
However, more comparisons are needed, specially mineral mass and vertical profile comparisons together with studies that consider the distinct mineral dust optical properties dependent on mineral composition.




## 5    Conclusions & Implications

An implementation of soil mineralogical composition into COSMO-MUSCAT's dust emission scheme is presented. The objective of the implementation is to be able to predict mineral mass concentration in the atmosphere. The simulated mineral mass is part of the mineral dust aerosol emitted from the Sahara Desert and transported towards the Atlantic Ocean. Special focus is given to the Cape Verde archipelago since a large array of measurements devices and approaches are available for validation purposes. The validations are presented for both a specific case of mineral dust plumes originating at the beginning of February 2022 and for the general spatial-temporal evolution of mineral dust mass concentrations in the atmosphere for January-February 2022. Most of the comparisons presented rely on an assumption of homogeneity for mineral dust optical properties.

The explicit representation of dust mineralogy in COSMO-MUSCAT is, to our knowledge, the first time that dust mineralogy is included in the set of parameterizations describing the mineral dust life cycle for a regional atmospheric model which opens the possibilities for comparing with specific field measurements. Comparisons with measurements show that the model represents the mineral content well. Nonetheless, the comparison with mineral mass concentrations showcases the lack of measurements and therefore opens up the possibility of validating mineral masses in other ways. It would be of particular interest to be able to use remote sensing methods to specify the effects of mineral composition, e.g, their iron content. By considering the amounts of iron oxides simulated in the atmosphere a link could be made with regards to changes on the optical retrieved properties for specific iron heavy simulated layers. If such a link is determined, a correlation between the amount of simulated iron oxide content and the UV/VIS response to it could be established (Balkanski et al., 2007; Veselovskii et al., 2020; Li et al., 2019; Zhang et al., 2015) and therefore, measured. Information that could be later be used for both monitoring mineral dust aerosols through lidar measurements and for creating a more accurate description of mineral dust aerosols and their coupling with radiative transfer processes on atmospheric models.

The regional chemistry transport model MUSCAT contains a radiative transfer feedback mechanism which could be further improved with mineral individual optical properties, depending on source regions and their specific mineral constitution. Considering the diversity of dust composition and their specific relation with the UV-VIS-IR wavelengths provides an excellent opportunity for further research into the direct radiative effect of dust. Such a inclusion has already been suggested by Balkanski et al. (2007) who already showed better matches with observations when varying mineral dust refractive indices by considering a fixed amount of iron oxides content. This suggests to us that varying mineral dust optical properties depending on their source regions and mineral content could only aid in improving predictions for the atmospheric radiative balance.

In summary, the comparisons with measurements presented show that the general spatial-temporal evolution comparison matches well with the simulated mineral dust mass concentrations. For instance, the temporal evolution and simulation based AOT calculations almost always resembles the AERONET station measured AOT in the studied period. The simulated mineral dust mass concentration vertical profile fits the lidar retrieved aerosol profile over Mindelo, São Vicente, Cape Verde generally but misses a near-surface portion of mineral dust due to a wider coverage of simulated precipitation and a lack of local emissions on the model.



In the technical aspect of the mineralogical implementation to the model emission scheme some recommendations can be considered in order to improve the accuracy of the mineral mass concentration predictions in the atmosphere. The largest modification could be the consideration of the change in particle size distribution from the mineral dust productive soil and the aerosol mineral composition following Perlwitz et al. (2015b, a). Another possible improvement is the consideration of

625     different densities between minerals, as iron oxides containing minerals, have densities that are twice that of other minerals (Perlwitz et al., 2015a). Nevertheless, validation through measurements are the only way to verify the effect of these changes. We stress again the need for more mineral mass concentration measurements.

Crucially, the implementation of mineralogical information into the dust emission scheme of COSMO-MUSCAT is a key element for the investigation of the relation of the lidar retrieved vertical profiles and mineral masses. Such a relation implies

630     a possible link between lidar measured optical properties and dust source regions. Since this relation is currently thought as an association between iron oxide content and lidar ratio, the focus of further modelling projects should be then the accurate representation of iron oxide content. In such a framework, internal mixtures of iron oxide content could also be considered in the mineralogical parametrization following previous modelling efforts such as in Perlwitz et al. (2015b, a).

*Code availability.* The dust emission code can be found at https://doi.org/10.5281/zenodo.8321174

635     Additionally a dust test case based on the Sahara Desert can be found at https://doi.org/10.5281/zenodo.8320600

*Data availability.* The aerosol optical properties retrieved from Polly[XT] signals are published under https://doi.org/10.5281/zenodo.8100298

## Appendix A: Measured mineral fraction data used for model comparison

| Sample time | Location | Size range | Minerals | Measured (%) | Simulated (%) |
|---|---|---|---|---|---|
| 29-July 2002 Alastuey et al. (2005) | Santa Cruz de Tenerife, Spain(28°19'N, 16°30'W) | Bulk | illite | 15 | 11.3 |
| | | | kaolinite | 14 | 6.7 |
| | | | calcite | 9 | 4.8 |
| | | | gypsum | 3.5 | 1.1 |
| | | | quartz | 23 | 45 |
| | Santa Cruz de Tenerife, Spain(28°28'N, 16°15'W) | Bulk | illite | 13 | 9.3 |
| | | | kaolinite | 10 | 7.5 |
| | | | calcite | 6 | 4.8 |
| | | | gypsum | 10 | 1.1 |



| | | | quartz | 23 | 46 |
|---|---|---|---|---|---|
| 13 Jan-13 Feb 2006<br>Formenti et al. (2008) | Banizoumbou, Niger<br>(13°30'N, 2°36'E) | Bulk | illite | 9.98±4.65 | 12.9 |
| | | | kaolinite | 46.34±5.1 | 8.5 |
| | | | calcite | 2.59±1.84 | 2.5 |
| | | | other feldspar | 2.32±1.03 | 16.2 |
| | | | quartz | 33.39±4.13 | 52.3 |
| 18-23 Jan 2008 -<br>28-31 Dec 2007<br>Jeong and Achterberg (2014) | São Vicente, Cape Verde<br>(16°51'N, 24°52'W) | Bulk | illite-smectite | 72.12±2.05 | 15<br>8.2[*] |
| | | | kaolinite | 6.5±2.74 | 5.25 |
| | | | calcite | 2.38±0.68 | 2 |
| | | | K-feldspar | 1±0 | 8.6 |
| | | | gypsum | 4.75±1.37 | 0.75 |
| | | | quartz | 8.88±2.05 | 59.1 |
| 13-23 July and 6-8 August 2005<br>Kandler et al. (2007) | Izana, Tenerife<br>(28°19'N, 16°30'W) | 1-2.5 μm | illite | 41 | 39.3 |
| | | | quartz | 8 | 9 |
| | | | calcite | 5 | 2.8 |
| | | | hematite | 1 | 1.1 |
| | | 10-20 μm | quartz | 8 | 72.5 |
| | | | calcite | 5.5 | 3.1 |
| | | | hematite | 0.5 | 1.1 |
| 13 May-7 July 2006<br>Kandler et al. (2009) | Tinfou, Morocco<br>(30°24'N, 5°6'W) | Clay | illite | 31.5 | 44.8 |
| | | | kaolinite | 6.26 | 16 |
| | | | quartz | 11.47 | 11.4 |
| | | | calcite-dolomite | 10.31 | 2.1 |
| | | | hematite | 1.88 | 1 |
| | | 2-20 μm | feldspar | 33.1 | 14.5 |
| | | | gypsum | 2.96 | 1.1 |
| | | | quartz | 15.3 | 80.5 |
| | | | calcite-dolomite | 11.72 | 2.9 |
| | | | hematite | 1.45 | 1 |
| 14 Jan-9 Feb 2008<br>Kandler et al. (2011) | Praia, Cape Verde<br>(14°56'N, 23°29'W) | Bulk | illite-smectite | 13.88±2.21 | 14.9 |
| | | | smectite | 6.09±1.06 | 8.2 |
| | | | kaolinite | 33.15±4.26 | 5.8 |
| | | | calcite | 2.18±0.41 | 2.2 |
| | | | K-feldspar | 20.65±2.38 | 9.2 |





| | | | gypsum | 4.37±1.47 | 1.1 |
|---|---|---|---|---|---|
| | | | quartz | 10.63±1.47 | 57 |
| Jan-Feb 1984-1985 Adedokun et al. (1989) | Ile-Ife, Nigeria (7°17'N, 4°20'E) | Bulk | kaolinite | 10.28 | 7.2 |
| | | | quartz | 74.78 | 54.5 |

Table A1: Mineral fraction data used in the model evaluation. The measurements were selected in basis of the dust plume origins. Where the measurements represent a whole period then, northern hemispheric winter was selected to match our simulation period. When measurements come from specific dust events then the comparison with the modelled data is done taking into account the dust plumes origins. *modelled results are reported in the following way, first illite, then smectite.

*Author contributions.* SGMA wrote the manuscript draft; DA and KS reviewed and edited the manuscript; DA and KS provided resources
such as study materials, instrumentation and analysis tools and were part of the conceptualization of the project; MF - software development
- restructured the code around MUSCAT dust emission scheme; SGMA, MF, BH, KS and IT contributed to code development surrounding
the mineralogy inclusion; SGMA, DA, HB, JH and AA were part of the formal analysis of lidar data. AS, BH, HB and RE are part of the
maintenance and continuous improvement of the Polly[XT] lidar device(s).

*Competing interests.* The authors declare that they have no conflict of interest

*Acknowledgements.* This study is done in the framework of the DUSTRISK (a risk index for health effects of mineral dust and associated
microbes) project, funded by the Leibniz Collaborative Excellence Programme Project (grant number K255/2019).

This research has been supported by the German Federal Ministry for Economic Affairs and Energy (BMWi) (grant no. 50EE1721C).
Furthermore, we also acknowledge the support through ACTRIS-2 under grant agreement no. 654109 from the European Union's Horizon
2020 research and innovation programme and ACTRIS PPP under the Horizon 2020 – Research and Innovation Framework Programme,
H2020-INFRADEV- 2016-2017, Grant Agreement number: 7395302.

Muñoz Sabater, J., (2019, 2021) was downloaded from the Copernicus Climate Change Service (C3S) Climate Data Store.

The results contain modified Copernicus Climate Change Service information 2020 and 2022. Neither the European Commission nor
ECMWF is responsible for any use that may be made of the Copernicus information or data it contains.

The AERDB D3 VIIRS NOAA20 Aerosol Optical Thickness 550 Land Ocean Mean daily L3 1x1 degrees grid datasets were acquired
from the Level-1 and Atmosphere Archive and Distribution System (LAADS) Distributed Active Archive Center (DAAC), located in the
Goddard Space Flight Center in Greenbel, Maryland.



Further thanks are due to the Deutscher Wetterdienst (DWD) for cooperation and support, to the developers of the LAGRANTO-COSMO tool and to all PIs of the AERONET stations used in this study for maintaining the instruments, obtaining the measurements and providing data.

We want to thank all the TROPOS team involved in the PollyNET, the network dedicated to provide continuous aerosol data from automated Raman-polarizations lidars (Baars et al., 2016) https://polly.tropos.de/.





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
