# Peer review of "The implementation of dust mineralogy in COSMO5.05-MUSCAT"

_EGUsphere, 2023_

## Author Comment (AC2)

**Replies to the comments from anonymous referee #1**

We would like to start the reply of the anonymous referee's comments by thanking them for their review and thoughtful revision of our manuscript. All the comments and insight are very much appreciated. We have copied their comments into this document; their comments are in Times New Roman blue font while our answers are in Calibri black font. Line numbers refer to the version of the manuscript with track changes.

This article describes the implementation of dust mineralogy in a regional model, COSMO5.05-MUSCAT, and presents an overall evaluation of the results. This implementation constitutes a first approach towards a more integrated representation of the dust mineralogy and its impacts, e.g., through the refinement of the definition of the optical properties. As such, the article addresses a relevant topic for the atmospheric and climate modeling communities and merits publication. However, in my view, some methodological aspects deserve a more detailed explanation and part of the highlights in the abstract and conclusions could be further clarified.

**General comments:**

- **Airborne minerals particle size distribution**

The authors assume the size distribution of the minerals reported in the soil mineralogy map of Nickovic et al. (2012) as equivalent to that in the airborne particles. Observational evidence (e.g., Kandler et al., 2009) show that phyllosilicates are often found in airborne dust in coarser sizes than those reported in the soil maps. The implications of that assumption for the size-resolved airborne mineralogy have been discussed in previous works (e.g., Perlwitz et al., 2015a,b, Pérez García-Pando et al., 2016, Gonçalves Ageitos et al., 2023) and a number of modeling studies include some form of adjustment between the soil mineral fractions and those in the aerosol (e.g., Scanza et al., 2015; Perlwitz et al., 2015a,b; Ito and Shi, 2016; Li et al., 2021; Gonçalves Ageitos et al., 2023). In my view, the authors should justify their choice to define the size distribution of the airborne minerals and further discuss its impact on their results throughout the article.

The reviewer accurately notes that measurements have previously indicated changes in the size distribution of minerals between soil parent and aerosol. Despite several authors incorporating these changes in previous modeling studies, we find it physically inconsistent to apply a similar method in COSMO-MUSCAT due to differences in the emission schemes.

Various studies (Scanza et al., 2015; Perlwitz et al., 2015a,b; Ito and Shi, 2016; Li et al., 2021; Gonçalves Ageitos et al., 2023) calculate mineral dust emissions and corresponding mineral mass emissions using the brittle fragmentation theory (BFT) proposed by Kok (2011). BFT proposes that energetic and repeated collisions like for soil aggregates mobilized by saltation, results in emitted aggregate diameters that are mostly smaller than a set scale.

The theory predicts particle emissions for diameters between approximately 2 and 20 μm. It describes the emitted particle size distribution as a power law. BFT assumes that the emission

of dust particles with sizes $D_d$ is proportional to the volume fraction of soil particles with sizes smaller than the dust particles diameters. The normalized emitted mass particle size distribution (PSD) can be expressed as:

$$\frac{dM_d}{dlnD_d} = \frac{D_d}{C_m}\left[1 + erf\left(\frac{\ln\left(\frac{D_d}{\overline{D}_s}\right)}{\sqrt{2}\ln(\sigma_s)}\right)\right]exp\left[-\left(\frac{D_d}{\lambda}\right)^3\right] \qquad (1)$$

Where $erf$ is an error function, and $C_m$ is a normalization constant. $\overline{D}_s$ and $\sigma_s$ are the volume median diameter and the geometric standard deviation from an invariant soil size distribution derived from measurements (Kok, 2011). With these values, a side crack propagation length, $\lambda$, can be calculated.

From this formula, it is possible to take into account the mass fraction of each mineral $i$ depending on its size, either defined only in clay, silt or both, and implement it into the mass PSD from BFT as:

$$\frac{dM_{di}}{dlnD_d} = \frac{m_{ci}D_d}{C_m}\left[1 + erf\left(\frac{\ln\left(\frac{D_d}{\overline{D}_s}\right)}{\sqrt{2}\ln(\sigma_s)}\right)\right]exp\left[-\left(\frac{D_d}{\lambda}\right)^3\right]; \; for \; D_d \le 2\mu m \qquad (2)$$

$$\frac{dM_{di}}{dlnD_d} = \frac{m_{ci}D_d}{C_m}\left[1 + erf\left(\frac{\ln\left(\frac{D_d}{\overline{D}_s}\right)}{\sqrt{2}\ln(\sigma_s)}\right)\right]exp\left[-\left(\frac{D_d}{\lambda}\right)^3\right] + \frac{m_{si}D_d}{C_m}\left[erf\left(\frac{\ln\left(\frac{D_d}{\overline{D}_s}\right)}{\sqrt{2}\ln(\sigma_s)}\right) - \right.$$

$$\left. erf\left(\frac{\ln(2/\overline{D}_s)}{\sqrt{2}\ln(\sigma_s)}\right)\right]exp\left[-\left(\frac{D_d}{\lambda}\right)^3\right]; for \; 2 < D_d \le 20\mu m \qquad (3)$$

Where $m_{ci}$ is the mineral mass fraction from mineral $i$ in clay size and $m_{si}$ the $i$ mineral mass fraction in silt size. The sum of all the mineral mass PSDs is set to be the same as the total dust mass PSD. Then, the emitted mass fraction of every mineral $M_{ik}$ in each size bin $k$ that are the atmospheric transport bins are calculated by integrating the mineral mass PSDs depending on the diameter limits established for each size bin as:

$$M_{ik} = \int_{D_{dkmin}}^{D_{dkmax}} \frac{dM_{di}}{dlnD_d} \frac{1}{D_d} \, dD_d \qquad (4)$$

Where the sum over mineral per size bin is set to 1 and $D_{dkmin}$ and $D_{dkmax}$ are the limits of each size bin.

In contrast, the emission scheme used in COSMO-MUSCAT follows the Marticorena and Bergametti (1995) emission scheme, where to define the mass PSD of dust is defined by considering all the size modes in the soil:

$$\frac{dM_d}{dlnD_d} = \sum_{j=1}^{n} \frac{M_j}{\sqrt{2}\ln(\sigma_j)} exp\left[\frac{(lnD_d - lnMMD_j)^2}{-2ln^2\sigma_j}\right] \qquad (5)$$

$j$ refers to the size mode, in our study, we consider three soil size modes, clay, silt and sand. $M_j$ is the mass fraction of particles for the mode $j$ and $MMD_j$ is the mass median diameter

and $\sigma_j$ is the geometric standard deviation of the $j$ mode. The equation expresses the soil mass size distribution for each particle diameter in terms of the sum of physical parameters of each soil mode. Therefore, each of the mass PSDs varying with particle diameter depend on the mass amount of clay, silt, and sand found in the soil.

Then a relative contribution to the total flux of each size range is assumed to be proportional to the relative surface it occupies on the total surface. Each surface covered by each grain is assimilated to its basal surface, and so, a PSD of the basal surfaces can be calculated via the mass PSDs by assuming sphericity and same density (homogeneity) as:

$$dS_d = \frac{dM_d}{\frac{2}{3}\rho_p D_p} \qquad (6)$$

From where a total basal surface $S_{total}$ can be obtained by summing over diameters of $dS_d$. After which a relative basal surface $dS_{drel}$ can be established by dividing the individual particle $dS_d$ by $S_{total}$. Finally, these relative particle basal surfaces can be used to define a horizontal flux distribution as a function of the particle diameter as:

$$dG_d = dS_{drel} * G_d \qquad (7)$$

Where $G_d$ is the horizontal flux that depends on the particle threshold friction velocity $U_t^*$, the overall friction velocity $U^*$, air density $\rho_a$, gravity and a constant $C$ of proportionality with a value set to 2.61 based on wind tunnel experiments (White 1979).

$$G_d = C \frac{\rho_a}{g} U^{*3} \left(1 + \frac{U_t^*}{U^*}\right)\left(1 - \frac{U_t^{*2}}{U^{*2}}\right) \qquad (8)$$

These horizontal fluxes are afterwards translated into vertical fluxes via an alpha coefficient which depends on the amount of clay, silt and sand found in the soil. Afterwards, depending on the particle's diameter, and the limits of the size bins, the relative mass PSDs multiplied by the fluxes are assigned to a dust bin that will conduct the transport of dust in the atmosphere.

The crucial difference and the reason why we chose not to use the mineral mass aerosol size distribution calculated by the above-mentioned authors using BFT is that the Marticorena and Bergametti (1995) scheme relies on the full soil characteristics per size mode in contrast with BFT that calculates a mass PSD based on a volume median diameter and the geometric standard deviation from an invariant soil size distribution.

Further work is currently ongoing that aims at being able to use the Marticorena and Bergametti (1995) dust emission scheme, including mineral masses in order to calculate the changes from the mineral's soil size distribution into aerosols.

Additionally, we would like to point that the current distribution of minerals in COSMO-MUSCAT partially considers that a change occurs in the mass PSD during the emission process and therefore only considers the mass fractions of the two smallest size modes (clay and silt).

We have expanded the justification throughout the "2.1 Mineralogy implementation" section. Major changes can be read between L211-225. We have added a paragraph in the "Discussion" section between L629-635, where we further discuss the impact of not considering the change in the mass particle size distribution when comparing the modelling results to mineral mass measurements.

- **Evaluation of the dust mineralogy**

The evaluation section presents a comparison of the modeled dust optical properties against different products and retrievals. This evaluation is relevant to prove the model's ability to represent the dust cycle (a necessary step to reproduce the dust mineralogy), but it could be shortened and/or included as supplementary information.

The focus should be put on the evaluation of the mineralogy, either through direct measurements or the use of mineralogy-sensitive optical properties (e.g., single scattering albedo). Furthermore, the mineral observations from in-situ data reported in Appendix A include information on aerosol samples with various size ranges, while the caption on figure 5 suggests that the scatterplots only consider bulk aerosol measurements. A size-collocated evaluation of the mineral fractions would increase the number of data available and provide relevant information on the ability of the model to reproduce the mineral content in different size ranges. Something that, on the other hand, could be linked to the assumed size distribution at emission (see the comment above).

Finally, the authors present a comparison of the dust vertical profile with LIDAR products, where they add the vertical profile of the modeled mineralogy. While they hypothesize that considering explicitly hematite would lead to better agreement with observations, this is not proved in their case study. I would suggest to clearly acknowledge this is the abstract and conclusions where they highlight this hypothesis.

We consider relevant to add the evaluation of the modeled dust optical properties against the different products and retrievals since this specific setup of COSMO-MUSCAT has not been used and therefore compared to measurements before.

Thank you for the suggestion of comparing the mineralogy by using mineralogy-sensitive optical properties such as the single scattering albedo, we will take it into account for further studies but we consider that such a comparison is beyond the scope of this manuscript for the following reasons:
  (1) Mineral specific refractive indices significantly vary between the literature found values [e.g., variation in hematite refractive indices as shown in Fig.3 from Go et al., (2022)]

(2) On the hypothetical of taking mean values for the mineral specific refractive indices, a broader sensitivity study would be granted. It is our opinion that such a study would be a good addition to the subject of our project, but beyond the purposes of this manuscript, which is the presentation of the introduction of mineralogy in COSMO-MUSCAT.

(3) Furthermore, the simulation period and region has as a consequence that the presence of other non-dust aerosols cannot be disregarded. Due to the state of the model, we cannot at the moment, account for this other aerosol species and their interaction with light. This would cause, in our opinion, the need to further investigate the effects of other aerosols in our measurements which leads us astray from the mineralogical study.

As to the point of taking into account the sampled sizes for the mineralogical comparisons, we indeed did the comparison with mineral observations by taking into account the differences in measured sizes. We have added the following sentence to elucidate this matter in the section "2.4 Observational data for model evaluation", at L298-301: "Furthermore, some measurements consider different size ranges. Whenever the measurements are reported as bulk size, the measurements are compared to the sum of the modelled bin sizes. Otherwise, the comparison is done taking into account the measurement size differences by being compared to the correspondent modelled size bin."

We have edited the caption of Fig.5, which now reads "Scatterplots of minerals mass percentages of illite, kaolinite, feldspar, calcite, hematite and quartz measured vs. simulated by COSMO-MUSCAT (see Table A1) The dashed lines represent the ratios of 2:1 and 1:2 between the simulated and observed mineral percentages. The error bars are present when reported in the measurements"

We consider that the abstract sentence regarding the hypothesis of hematite content leading to a specific feature on the LIDAR vertical profiles clearly states that this is not proven in the case study due to the wording "highlighting the possibility". Moreover, we consider that the acknowledgment of the hypothesis not being proven in the case study is sufficiently done in the paragraph found between L557-559 in the "Comparison to lidar remote sensing and model-base attribution of local mineralogical properties" section. The conclusion, in our view, does not highlight the case study, rather, the hypothesis.

- **Mixing state of the iron oxides and mass density**

COMO-MUSCAT uses a representation of the different minerals as external mixtures. In the case of iron oxides, previous works suggest that these minerals are often found as accretions in other mineral particles (e.g., Kandler et al., 2009). Iron oxides have greater mass densities than other dust mineral components, which would make their lifetime in the atmosphere shorter. The methods section should clarify which is the assumed mass density for the different minerals, and particularly for the iron oxides. There are a couple of remarks in the conclusions

related to this aspect, but it is unclear to the reader how this was dealt with in COSMO-MUSCAT.

As the reviewer accurately points out, works such as Kandler et al., 2009 suggest that iron oxides affect the density of mineral dust. The physical characteristics of mineral dust, such as density, in COSMO-MUSCAT are treated as bulk characteristics due to the nature of the emission scheme, as seen in equations 5 and 6. Besides, the particle threshold friction velocity $U_t^*$ used for the calculation of the emissions fluxes (Eq. 8) depends on the particle density. Changing the density considering changes in composition would therefore need a mineral specific treatment which is not yet implemented for a similar reason as for the changes in mass particle size distributions.

Additionally, we have run sensitivity test where we change the assumed bulk mineral dust density from 2650 kg/m$^3$ to 2710 kg/m$^3$ considering the highest amount possible of hematite and goethite found in the Sahara and the impact on the emission flux is on average -0.5%. We have added a sentence in the section "Mineralogy implementation" clarifying the assumed mass density for mineral dust together with a justification at L240-244.

**Specific comments**

L8,9 - How is the improvement of model performance related to the introduction of the mineralogy?

We have rephrased that sentence, since the mineralogy does not lead to the improvement of the model performance but the specific set of physical parametrizations adjusted in the emission scheme does. The new sentences are "We provide a detailed description of the implementation of the mineralogical database, GMINER (Nickovic et al., 2012). Together with a specific set of physical parametrizations in the model's mineral dust emission module which lead to a general improvement of the model performance when comparing the simulated mineral dust aerosols with measurements over the Sahara Desert region for January - February 2022." (L7-10)

L13,15 - Please, clarify how your results back up the link between hematite and the improved interpretation of the LIDAR product.

There is no information on the sentence stating that there is an interpretation improvement. Furthermore, we state that the results highlight a possibility of linkage between the model with resolved mineralogy and the interpretation of lidar measurements.

L16 - Why does the comparison with in-situ measurements show how important they are?

We have removed that part of the sentence (L16-17)

L27 - Jickells et al. (2005) focuses on dust as a source of iron for marine ecosystems, rather than its direct radiative effect.

We have removed that reference (L28)

L29 - Chatziparaschos et al. (2023) is a modeling study. It focuses on k-feldspar and quartz as INPs. One could cite Harrison et al., 2019 as an observationally based study:

*Harrison, A. D., Lever, K., Sanchez-Marroquin, A., Holden, M. A., Whale, T. F., Tarn, M. D., Mcquaid, J. B., & Murray, B. J. (2019). The ice-nucleating ability of quartz immersed in water and its atmospheric importance compared to K-feldspar. Atmos. Chem. Phys, 19, 11343–11361. https://doi.org/10.5194/acp-19-11343-2019.*

We have removed Chatziparaschos et al. (2023) and added Harrison et al. (2019). Thank you for the suggestion.

L44,46 - Specify the source of the soil types information as FAO classification. Nickovic et al. (2012) did not include additional mineralogy measurements, but added the phosphorus content of the soils (an element present in different minerals). Please, rephrase.

We have rephrased in order to include the FAO classification and that Nickovic et al. (2012) added phosphorus and further soil types into the classification. The sentences are now (L45-49) "Claquin et al. (1999) proposed that the soil mineral fractions are approximately related to the soil type; taking into account the size distribution, the chemistry and the color of the soil according to the FAO74 classification (FAO-UNESCO, 1974). They derived an average surface mineralogy that can be inferred for each soil unit of the arid soil. Nickovic et al. (2012) extended this approach by including new soil types and phosphorus concentrations."

L55 - There are previous works implementing mineralogy in regional models:

*Menut, L., Siour, G., Bessagnet, B., Couvidat, F., Journet, E., Balkanski, Y., & Desboeufs, K. (2020). Modelling the mineralogical composition and solubility of mineral dust in the Mediterranean area with CHIMERE 2017r4. Geosci. Model Dev, 13(4), 2051–2071. https://doi.org/10.5194/gmd-13-2051-2020*

Thank you for letting us know. We have changed the sentence and acknowledge them (L57 - 59)

L56 - There's something missing in the sentence.

Rephrased (now L60)

L66,77 - There are multiple modeling exercises that consider a different mineral particle size distribution in the aerosol than in the soil. Some examples are: Scanza et al. (2015), Li et al. (2021) for the CAM model, Gonçalves Ageitos et al. (2023) for MONARCH, Chatziparaschos et al. (2023) for TM4, Myriokefalitakis et al. (2021) for EC-Earth3, Ito et al. (2016) for IMPACT, etc.

The main idea of this paragraph is not to point towards modelling studies that have considered the differences between soil mineral mass particle size distribution and aerosol

mineral mass particle size distribution, rather the objectives of different modelling efforts that have included mineralogy, independent of their size distribution treatment. Perlwitz et al. (2015a,b) is cited because of the paper's objective on predicting the regional variations by explicitly considering the change on the mineral mass particle size distributions. We want to stress the variety of studies for which the implementation of mineralogy can be useful.

L78 - I would recommend stating the main objectives of the paper before describing the content of each section.

We have added a sentence to this regard, thank you for the suggestion. L84-86: "This paper aims at describing and validating both the implementation of mineralogy and the set of physical parametrizations used to simulate the atmospheric life cycle of mineral dust. Ways in which further studies could benefit from a model with resolved mineralogy are highlighted throughout the comparisons with dust related measurements."

L176 - "Effective fractions of minerals in soils are determined by combining soil texture classes and applying modifications derived from modelling approaches." This is not clear. Please, clarify.

This sentence became "Effective fractions of minerals in soils are determined by combining soil texture classes." (L185)

L179 - Phosphorous is not a mineral, and as far as I understand it has not been included in COSMO-MUSCAT. Please, clarify.

Phosphorus has been added to COSMO-MUSCAT, but the phrasing pointed towards phosphorus being a mineral and as the reviewer correctly points out, it is not. The sentences between L186-190 changed to "GMINER is consequently a database that establishes the relationship between different mineral dust-productive soil types and the following minerals: quartz, feldspar, calcite, gypsum, illite, kaolinite, smectite and hematite. Phosphorus, which content is found in several minerals and which concentration is crucial for its role in ocean fertilization was also added. Mineral and phosphorus fractions are distributed over clay and silt particle size population,…"

L186,187 - See my general comment above. This assumption must be justified in view of previous evidence and or its implications discussed more thoroughly in the article.

We have added more information for the justification and discussed the impacts on the results later on (L211-225& L629-635).

L191,193 - How does the GMINER dataset consider the mineralogical composition changes during the emission process? "By only taking into the account the soil mineralogical composition of the particle classes that would be emitted, that being, silt and clay sizes." This sentence is incomplete. Please, if it refers to the previous sentence, rephrase and clarify. The fact that the soil mineralogy encompasses the clay and silt size ranges does not mean that the

size distribution within these particle sizes measured in wet sieved soils would correspond to that of the emitted aerosol.

It is true that by taking into account the mineralogical fractions of silt and clay there is no consideration on how the mineral mass particle size distribution would change between soil and aerosol. We have rephrased the sentence to clarify that GMINER already implicitly considers a size distribution of what could become an aerosol but not the change that that size distribution would undergo during the emission process (L200-205).

L199 - "That causes a larger allocation of mineral fractions to clay sized populations than could exist in undisturbed soil." Please, clarify if this is the case for all minerals. Also, refer to the potential implications of this in your modeling study.

The implication towards the modeling of phyllosilicates was added to this sentence (L211-213).

L205,206 - "Results were later supported by Perlwitz et al. (2015b) who found that the particle composition of the clay sized emitted minerals is identical to that of the fully dispersed soil as given by Claquin et al. (1999)." Please, review this sentence. Perlwitz et al. (2015) found that reaggregation before emission improves the modelled aerosol mineral fractions below 2 um (e.g., for feldspars).

Perlwitz et al. (2015) found that the emission of phyllosilicates is very similar to the soil parent distribution of phyllosilicates but this is after considering the reaggregation coefficient which produces phyllosilicates content at silt sizes as well and not only in clay sizes as proposed by Claquin et al. (1999). Consequently, our sentence has been removed.

L213,214 - The mentioned works do not present an evaluation of the size distributed mineralogy of airborne dust particles against mineralogy measurements. Atkinson et al. (2013) and Journet et al. (2014) do not show any evaluation of the modeled mineralogy. Hoose et al. (2008) shows an evaluation of phyllosilicate mass fractions (i.e. kaolinite, illite/smectite) without considering the particle sizes and the modeled results are not particularly correlated with the observations.

True, they do not present an evaluation of the size distributed mineralogy, but the idea behind this sentence is to point towards other studies which have used a similar approach as ours. Therefore, we chose to keep it.

L231 - It has been shown that iron oxides (e.g., hematite) are usually internally mixed with other minerals (in the form of accretions in the surface of the mineral). This mixing has implications in terms of the transport (i.e., the hematite particles are much denser than the average dust particles, therefore if transported as externally mixed, they would be removed from the atmosphere efficiently close to sources). How are these aspects treated in COSMO-MUSCAT? Discuss how the external mixing assumption could affect the results shown.

This is now discussed when the density consideration is introduced at L240-244.

L240 - The evaluation of the modeled AOT against AERONET retrievals provides relevant information on the COSMO-MUSCAT ability to reproduce the dust cycle (see my general comment above). The authors could consider complementing this evaluation with single scattering albedo (SSA) from AERONET, which is sensitive to the mineralogical composition. Besides the selection of stations close to dust sources, other AERONET parameters could be used to identify retrievals dominated by dust.

We appreciate the suggestion of including SSA and would consider it for further works. The Ångström exponent for the 440-870nm wavelength range has been added to the AOT plot from AERONET in order to give clarity to the claim that most of the AOT retrieved is caused by mineral dust.

L279,280 - Please, specify the link between the LIDAR data and the mineral resolved emissions of dust.

The sentence was rephrased to: "The advantage of the comparison with lidar data is that it can indicate two things implicitly in a positive comparison: first, it can be used to confirm the simulated data, and second, some measured lidar data sets can hint to different Saharan origins by linking a specific feature in the UV-VIS signals in lidar measurements to the modelled UV absorbing minerals concentrations (Veselovskii et al., 2020)." (L312-315).

L408 - Please, see my general comment above. I would recommend adding to this comparison the size-dependent mineralogy evaluation. Also, if possible, I would include the observations in the x-axis and the model in the y-axis in Figure 5. The interpretation of the figure would be then clearer with values above the 1:1 line representing an overestimation, and below, an underestimation.

We have changed the figure axis as suggested and removed the confusion about the size dependency of the evaluation.

L412 - I would recommend adding some quantitative metric to back up the assessment, e.g. "good agreement with measurements". The number of points used for the evaluation is also relevant when interpreting the results.

We have accordingly changed the assessment of the figure at L450-453.

L513 - See my general comment above.

We consider that the final sentence of the paragraph acknowledges that the hypothesis is not proven in this study case (L557-559).

L569,574 - Balkanski et al. (2007) focuses on the dust absorption, rather than the evaluation of the AOT.

That sentence is now rephrased in order to clarify that Balkanski et al. (2007) does not evaluate the AOT. It now reads "For example, the study of Balkanski et al. (2007) that focused on the evaluation of dust radiative forcing, suggested that the reason for model over

estimations of aerosol mineral dust lay on the discrepancy on mineral shortwave refractive indices. Their study shows, how, by modifying the homogeneously assumed optical properties of mineral dust, better agreements with AERONET measurements can be found." (L614-616).

L576 - Please, clarify why the measured quartz and feldspars are less reliable than other minerals' content.

An example was given regarding why no assessment can be done with the feldspar comparison and the paragraph was rephrased to include the quartz findings (L621-623).

L596,597 - See my comment above (L55).

Changed accordingly. The sentence has changed to "The explicit representation of dust mineralogy in COSMO-MUSCAT is part of a handful of studies (Solomos et al., 2023; Menu et al., 2020) that include dust mineralogy in the set of parameterizations describing the mineral dust life cycle in a regional atmospheric model which opens the possibilities for comparing with specific field measurements" (L649-652).

L612 - See my comments above (L240 and L569-574).

Balkanski et al. (2007) study is highlighted as a dust radiative effect study in this sentence (L665-667).

L622,625 - See my comments above. These aspects must be clarified and justified in the methods section.

A clarification has been added in the methodology section (L211-225).

**Technical corrections**

There are some acronyms in the text that are not defined the first time they appear. I would also recommend the authors to review that the references are appropriately included (and review the format in the bibliography).

We have checked that and corrected when necessary. Thank you

References

[revised manuscript text omitted]

---

## Author Comment (AC3)

**Replies to the comments from anonymous referee #2**

First of all, we would like to thank the reviewer for their thorough revision of our manuscript. All the comments and insight are very much appreciated. We have copied their comments into this document; their comments are in Times New Roman blue font while our answers are in Calibri black font. Line numbers refer to the version of the manuscript with track changes.

The authors present a study describing the implementation of dust mineralogy in COSMO5.05-MUSCAT regional model. The results are compared with lidar, satellite, AERONET measurements and dust composition data from literature. This study is particularly relevant because there aren't many models that consider the dust mineralogy. I have a concern with respect to methodology and a few other points that should be addressed before publication:

1. In the abstract, introduction and conclusion, the authors wrote that this study is the first implementation of explicit representation of dust mineralogy in regional model. However, there is at least one older reference (Menut et al., 2020) dealing with this topic in regional model. Please add the reference.

Thank you for the clarification. We have changed that and added another addition of mineralogy in a regional model: Solomos et al., 2023

*Solomos, S., Spyrou, C., Barreto, A., Rodríguez, S., González, Y., Neophytou, M. K. A., Mouzourides, P., Bartsotas, N. S., Kalogeri, C., Nickovic, S., Vukovic Vimic, A., Vujadinovic Mandic, M., Pejanovic, G., Cvetkovic, B., Amiridis, V., Sykioti, O., Gkikas, A., and Zerefos, C.: The Development of METAL-WRF Regional Model for the Description of Dust Mineralogy in the Atmosphere, Atmosphere, 14, 1615, https://doi.org/10.3390/atmos14111615, 2023.*

2. Even if the simulations are compared with measurements and show relatively good results, how to know if the total amount of dust is conserved with mineralogy and with no mineralogy description? Is it possible to add a reference simulation with no mineralogy to estimate the potential benefit of this development?

Fig.1 illustrates an example that showcases how the total amount of dust (black line) is closely related to the the sum of all the minerals (green line). We expect sometimes to have less mineral mass than total dust mass due to the lack of mineral information in some regions. Furthermore, we consider that the method itself shows how the mass

conservation of dust is ensured. Therefore, we do not consider it necessary to include a reference simulation with no mineralogy.

[Figure]

Figure 1. Surface dust mass concentration at Mindelo, Cabo Verde grid cell for 10 January 01:00 UTC – 12 January 00:00 UTC 2022. Black curve represents the total dust mass concentration, green curve represents the sum of each mineral mass concentration. The other curves show some examples of selected minerals mass concentrations where brick red curve represents the quartz mass concentration, red curve represents illite mass concentration and blue curve shows kaolinite mass concentration.

3. I do not understand why you could consider the same relative part of silt and clay from the soil to the aerosol. There is a lot of literature available to take into account the relative part of clay and silt from soil to aerosol (eg: Scanza et al., 2015 simplified in Menut et al., 2020, Gonçalves Ageitos et al., 2023…)

We have answered a similar concern on our reply to anonymous referee #1, please refer to our answer to the first comment. Furthermore, we have made changes in order to clarify and justify our decision throughout the "2.1 Mineralogy Implementation" section, where major changes can be read at L211-225.

4. In the Methodology you discuss model configuration and emissions, but you don't mention deposition processes? How are minerals deposited?

The deposition processes are mentioned in L120 – 123.

5. Finally, you used the same density for each mineral and you used a fixed composition to calculate $Q_{ext, 500nm}$ for each size class. How far is this fixed composition from the simulated one? Can you explain why do you choose to do that? It would seem relatively easy to run sensitivity tests taking into account the simulated composition range.

Thank you for your recommendation. We would like to consider the changes on the $Q_{ext,550nm}$ parameter due to changes in composition, and to implement such changes in the model's radiation scheme. We consider that such a study is not trivial since the mineral specific optical properties found in literature vary in significant ways (Go et al., 2022). Furthermore, such an addition to our study is outside the scope of this

manuscript, where we focus on the addition of the mineral soil map to the emission scheme. The impact of mineral dust compositional changes in the atmospheric radiation balance is definitely a project we are interested in pursuing.

*Go, S., Lyapustin, A., Schuster, G. L., Choi, M., Ginoux, P., Chin, M., Kalashnikova, O., Dubovik, O., Kim, J., da Silva, A., Holben, B., and Reid, J. S.: Inferring iron-oxide species content in atmospheric mineral dust from DSCOVR EPIC observations, Atmos. Chem. Phys., 22, 1395–1423, https://doi.org/10.5194/acp-22-1395-2022, 2022.*

6. For the validation, you used only 5 AERONET stations. In fact there are more in your simulation area. Why don't you use all available data? How do you choose your stations? Can you plot the Ångström coefficient to be sure they're mostly dust?

We chose the AERONET stations that are on the dust path towards the Atlantic Ocean and that were actively measuring and provided cloud-screened data (level 1.5 or 2.0) for the studied period (Jan-Feb 2022). We appreciate the suggestion of adding the Ångström Exponent and we have consequently added it.

7. The validation is almost complete (lidar, satellite, AERONET measurements and dust composition from literature). Only mass concentration at the ground is not use. It could be interesting to use INDAAF data (https://indaaf.obs-mip.fr/catalogue) to compare your simulation with the $PM_{10}$

Even though we appreciate the comment and the idea of validation with INDAAF data, we consider it at the moment, outside the scope of the study, since we would want to focus on the mineralogical validation. We consider that the confirmation of COSMO-MUSCAT's ability of representing dust atmospheric life cycle is well covered with the cases presented.

8. What is the cost in computation time of implementing this level of detail?

The implementation of minerals in MUSCAT creates 60 additional tracers from which emission and transport is calculated. This addition results in an increase of roughly 10 times of the computational time used for these two processes. In total, the mineral implementation translates into an increment of computational time for the whole model system, COSMO-MUSCAT, of 67%.

9. Can the mineralogy representation be used with activated chemistry? Are there any impacts of this representation on heterogeneous reaction?

COSMO-MUSCAT current state cannot consider mineral dust as a chemically active aerosol. We could hypothesize that if it was allowed to interact in heterogeneous reactions that this would impact its optical properties, nevertheless, this is not being considered as part of the project for the time being, even though the concept is really interesting and we will revisit it again for future works.